

# Nacelle power curve measurement with spinner anemometer and uncertainty evaluation

Giorgio Demurtas[1], Troels Friis Pedersen[1], and Rozenn Wagner[1]

[1]DTU Wind Energy, Frederiksborgvej 399, 4000 Roskilde

*Correspondence to:* Giorgio Demurtas (giod@dtu.dk)

**Abstract.** The objective of this investigation was to verify the feasibility of using the spinner anemometer calibration and nacelle transfer function determined on one reference turbine, to assess the power performance of a second identical turbine. An experiment was set up with a met-mast in a position suitable to measure the power curve of the two wind turbines, both equipped with a spinner anemometer. An IEC 61400-12-1 compliant power curve was then measured for both turbines using
the met-mast. The NTF (Nacelle Transfer Function) was measured on the reference turbine and then applied to both turbines to calculate the free wind speed. For each of the two wind turbines, the power curve (PC) was measured with the met-mast and the nacelle power curve (NPC) with the spinner anemometer. Four power curves (two PC and two NPC) were compared in terms of AEP (Annual Energy Production) for a Rayleigh wind speed probability distribution. For each turbine, the NPC agreed with the corresponding PC within 0.10% of AEP for the reference turbine and within 0,38% for the second turbine, for
a mean wind speed of 8 m/s.

Keywords: Nacelle power curve, NPC, spinner anemometer, Nacelle transfer function, NTF

## 1   Introduction

Measuring the performance of a wind turbine means establishing the relation between wind speed (input) and electric power
(output). While the measurement of the electric power is straight forward (because it is already in electrical form), the challenge is to measure the wind speed. The IEC61400-12-1 standard describes the instrumentation requirements and the calculation procedures to determine the power curve with the method of bins, measuring the wind at hub height upstream of the wind turbine with a cup anemometer installed on a meteorological mast. A met-mast is costly, therefore the IEC61400-12-2 standard was developed to define requirements and procedures to measure the wind speed on the wind turbine. While the use of the
nacelle anemometer (mounted on the nacelle roof) for performance measurements is a well established procedure, the spinner anemometer is a less experienced option to measure the wind turbine performance. A spinner anemometer (Pedersen (2007)) consist of three one dimensional sonic wind speed sensors mounted on the spinner of the wind turbine. The advantage of a spinner anemometer over a nacelle anemometer is that it is measuring in front of the rotor rather than behind, where the flow is influenced by the wake of the blades and other elements present on the nacelle as described by Frandsen et al. (2009).



The spinner anemometer must be traceable calibrated using a met-mast in order to measure the wind speed accurately and to obtain an absolute power curve, according to the standard IEC61400-12-2 (2013) and as reported by Demurtas (2014).

Installation of a met-mast for each wind turbine is obviously not viable. Therefore the possibility of using the calibration found on a first -reference- turbine with a spinner anemometer to another one of same type was investigated in this work.

5      The objectives of the investigation were to:

– Install a met-mast to measure the power curve (PC) on two wind turbines next to each other.

– Install spinner anemometer on both wind turbines.

– Calibrate the spinner anemometer on the reference wind turbine.

– Measure the nacelle transfer function (NTF) on the reference wind turbine.

10      – Compute the NPC and PC for the reference turbine

– Apply the calibration values and NTF measured on the reference turbine to the second turbine.

– Compute the NPC and PC for the second turbine

– Compare the NPC with PC for both turbines.

– Evaluate the uncertainty related to spinner anemometer measurements



## 2   Site description

The measurements were taken at the Nørrekær Enge wind farm, located in the north of Denmark. This wind farm consist of a row of 13 Siemens 2.3 MW wind turbines (Fig. 1) in a very flat site, 80 m hub height and 93 m in rotor diameter. Every wind turbine was equipped with a spinner anemometer, but only the data from turbine T4 and T5 were used in this work. For this

5   experiment, an IEC61400-12-1 (2005) compliant met-mast was erected near turbine T4 and T5 (Fig. 2).

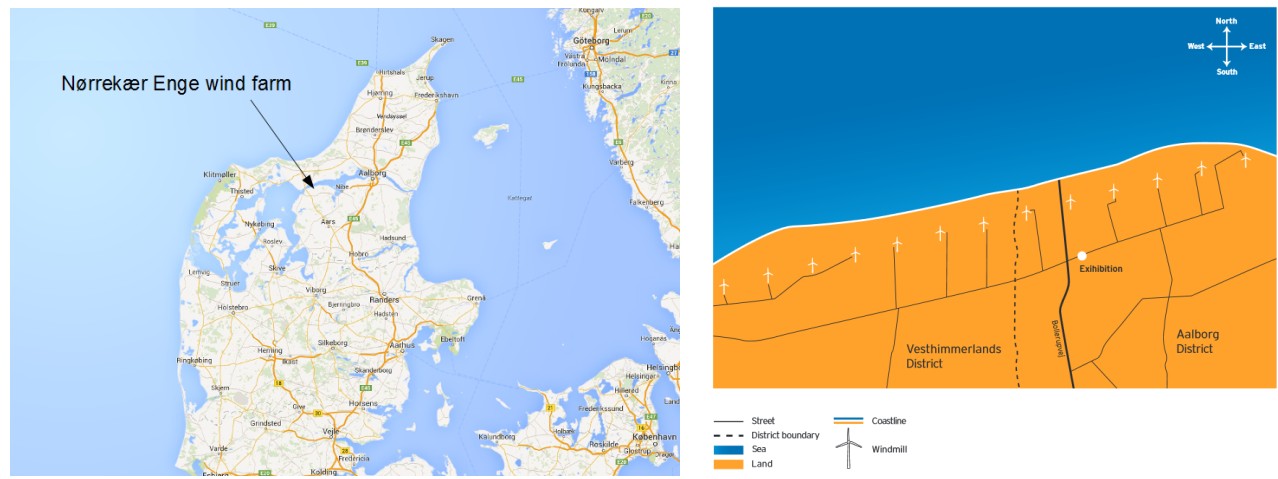

**Figure 1.** Left: Location of the wind farm in Denmark. Right: location of the 13 wind turbines in the wind-farm. The turbines are numbered 1 to 13 from the left to the right.

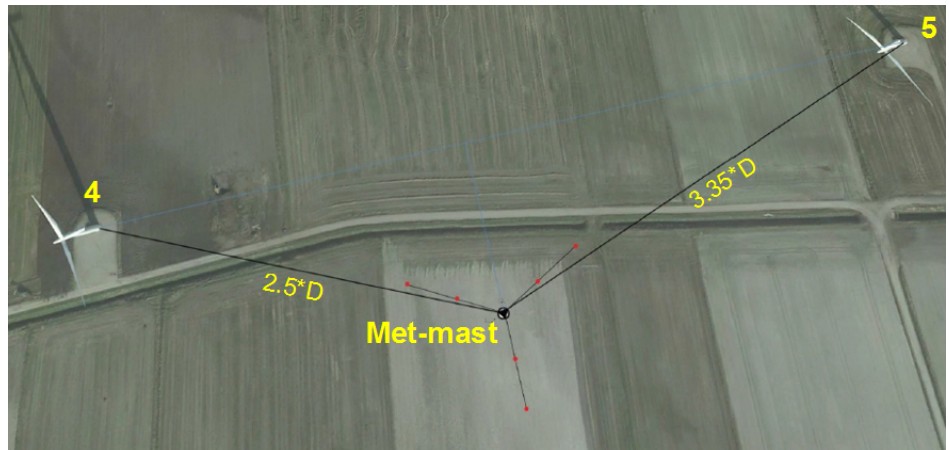

**Figure 2.** Relative position between reference turbine T4, met-mast and turbine T5.



The met-mast was positioned 2.5 rotor diameters from turbine T4 and 3.35 rotor diameters from turbine T5 (Fig. 2). The met-mast was equipped with a top mounted cup anemometer at 80 m a.g.l. (above ground level) at hub height, a wind vane at 78 m , barometer, thermometer and hygrometer at 78 m a.g.l.

The met-mast used a data-logger for meteorological measurements connected with a 3G modem to a server of DTU Wind
Energy. In each turbine the spinner anemometer was connected to a local data-logger and with a 3G modem to a server of Romo Wind A/S. The electric power produced by the wind turbine was measured with additional voltage and current transducers and the same data logger used for the spinner anemometer (for more details see Demurtas (2015)).

## 3   Spinner anemometer calibration

Calibration of spinner anemometer has been analyzed and investigated in Pedersen et al. (2015) and Demurtas et al. (2016).
Due to the large size of the spinner of a modern wind turbine it is not feasible to place it directly into a wind tunnel. Therefore each sonic sensor was first calibrated in the wind tunnel, and then, once mounted on the spinner, internally calibrated (for details see the manual of the spinner anemometer by Metek (05-01-2009)). The internal calibration procedure ensures that the three sensors read the same average wind speed. The spinner anemometer on T4 was $k_\alpha$ calibrated to ensure that the inflow angle is measured correctly, and $k_1$ calibrated to ensure that the output value $U_{hor}$ equals the free wind speed when the turbine
is stopped and pointed to the wind (see Demurtas (2014) for details). The $k_\alpha$ and $k_1$ calibration values found for T4 were used on both T4 and T5 (which is reasonable as long as the mounting of the sonic sensors and the spinner shapes are equal).

### 3.1   Sonic sensors wind tunnel calibration

The objective of the calibration of individual sonic sensors is to calibrate the wind speed measurements by the sonic sensors $V_1$, $V_2$ and $V_3$. Each sensor was calibrated individually in a MEASNET compliant wind tunnel. The sensor was mounted on a
support plate to hold it in the wind tunnel test section, Fig. 3. The mounting plate geometry was defined in Demurtas (2014) and the procedure described in IECRE (2015)). A calibration certificate was released for each sonic sensor. The values resulting from the wind tunnel calibration (slope $m$, offset $q$ and sensor path angle $\phi_s$, Tab. 1) should be set in the spinner anemometer conversion box (which converts $V_1$, $V_2$ , $V_3$ and the rotor position into $U_{hor}$, $\gamma$ and $\beta$) with the method described in Demurtas (2014). However this was not done, and a correction was applied to the measurements afterwards (see section 4). The sensor
path angle $\phi$ was not used.




**Table 1.** Sensor path angle ($\phi_s$), slope ($m$) and offset ($q$) coefficients of the sonic sensor wind tunnel calibrations.

|          | Turbine 4 (SN: 107114721) | | | Turbine 5 (SN: 107114722) | | |
|----------|---------|---------|--------------|---------|----------|--------------|
|          | $m$     | $q$     | $\phi_s$     | $m$     | $q$      | $\phi_s$     |
| Sensor 1 | 1.20746 | 0.18431 | 34.7°        | 1.22198 | 0.07906  | 34.7°        |
| Sensor 2 | 1.22794 | 0.00168 | 34.8°        | 1.23066 | -0.08116 | 34.6°        |
| Sensor 3 | 1.23249 | 0.16930 | 35°          | 1.21517 | -0.56490 | 34.1°        |
| Average  | 1.22263 | 0.11843 | 34.7°        | 1.22198 | 0.07906  | 34.7°        |

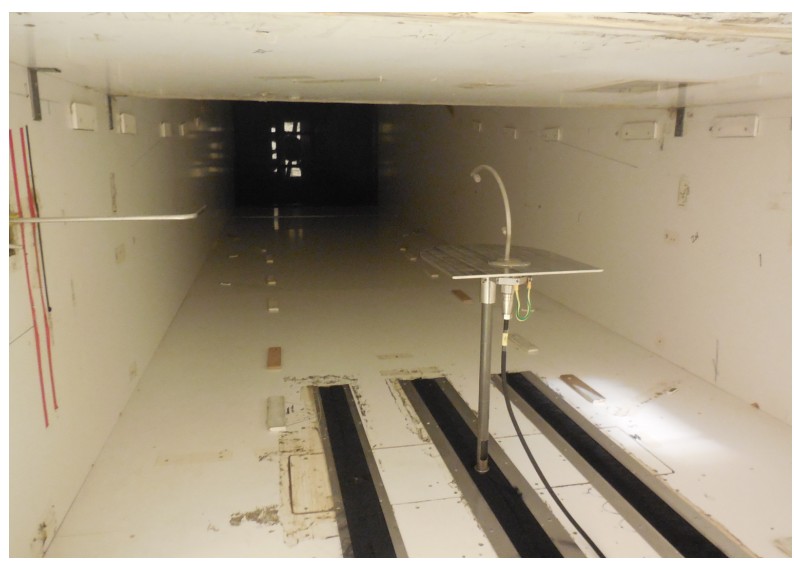

**Figure 3.** One sonic sensor mounted on the mounting plate in the test section of the SOHansen wind tunnel. The reference pitot tube is visible at the left hand side of the photo.

## 3.2 $k_\alpha$ calibration

The calibration for inflow angle measurements was made with the wind speed response method (WSR) described in Demurtas and Janssen (2016). The turbine was yawed several times of plus minus $60°$ in good wind conditions. The resulting calibration value $k_\alpha = 1.442$ was used to correct the measurements with the procedure described in Pedersen et al. (2015). The uncertainty on the $k_\alpha$ value could be calculated by repeating the test several times (as was done in Demurtas and Janssen (2016), which found a repeatability of the result within $8.5\%$ of the mean value, for a different wind turbine model). In this case the calibration test was performed only once, and the uncertainty was estimated to $u_{k\alpha} = 10\% \cdot k_\alpha$.





### 3.3  $k_1$ calibration

The objective of this calibration is to find the value of the $k_1$ calibration constant that makes $U_{hor}$ to match the free wind speed $U_{mm}$ when the turbine is stopped and is facing the wind. During operation of the wind turbine the rotor induction is accounted for with the nacelle transfer function (NTF) as described in the IEC61400-12-2 standard. To acquire the measurements needed for the calibration the wind turbine should be stopped, so that the wind seen by the spinner anemometer is not influenced by the induction. However, stopping the wind turbine would cause an energy loss, therefore the calibration was performed with the wind turbine in operation at high wind speed as proposed by Demurtas et al. (2016).

The $k_1$ calibration procedure was based on measurements acquired during operation of the wind turbine where $k_1$ was set to the default value $k_{1,d} = 1$ in the spinner anemometer conversion box. The correction factor $F_1$ was calculated as the ratio

$$F_1 = \frac{U_{hor,d,c}}{U_{mm}} \tag{1}$$

where $U_{hor,d,c}$ is the horizontal wind speed measured with default $k_{1,d}$ and calibrated $k_\alpha$.

Since T4 is pitch regulated, $F_1$ should tend to an asymptote as the wind speed increase (Fig. 4) because the induction decreases for high wind speed. The value of $F_1 = 0.6019$ was calculated as the average of the values for free wind speed greater than 15 m/s. Since the default value was $k_{1,d} = 1$, the calibration value is:

$$k_1 = F_1 \cdot k_{1,d} = 0.6019 \tag{2}$$

$k_1$ is not subject to uncertainty because it is compensated with the uncertainty estimation of the NTF. This is further explained in section 9.

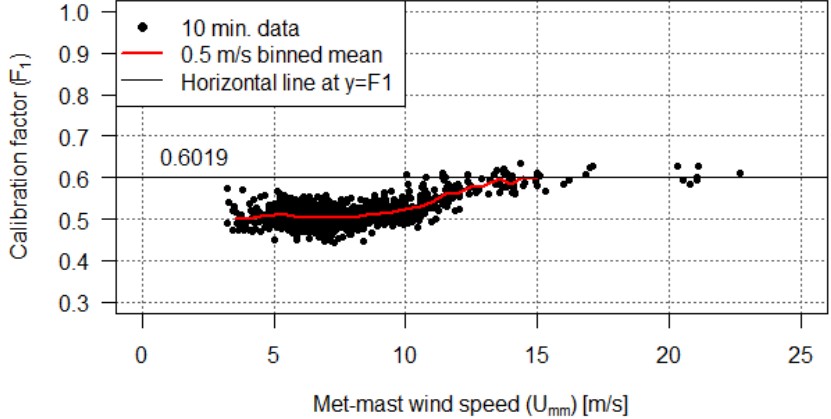

**Figure 4.** Calibration factor $F_1$ as a function of free wind speed during operation of the wind turbine.



## 4  Measurement database, data filtering and corrections

The measurement data-base consists of 237 hours of measurements acquired in a free wind direction sector between $101°$ and $229°$ as measured by the wind vane on the met-mast. The spinner anemometer measurements from both turbine 4 and 5 were calibrated with the $k_\alpha$ and $k_1$ values found for T4. Ten minute data sets where the minimum wind turbine rotor rotational speed

was lower than 40 rpm were filtered out in order to keep data where the turbine is continuously in operation. Data sets where the ten minute mean power coefficient (measured with the met-mast) were higher than 16/27 (the Betz limit) were filtered out to remove four outliers (this is a deviation to the requirements of the IEC61400-12-1 (2005) standard). There was no need to filter for freezing temperature, since the temperature was between 6 and $14°$C.

The sonic sensors wind tunnel calibration values were not set in the spinner anemometer conversion box as required in

Demurtas (2014). However a correction was made on the measurements to take into account the results of the wind tunnel calibration. From the calibration certificates the sensors on turbine 5 has smaller slope coefficient ($m_5$) and smaller offset ($q_5$) than those on T4 ($m_4$ and $q_4$), which means that the sensors on T5 are reading a bit higher wind speed than sensors on T4. Measurements of T5 were reduced with the ratio of the mean slope and the difference in mean offset.

$$U_5 = U_{5,original} \cdot (m_5/m_4) + q_5 - q_4 \tag{3}$$

Figure 5 shows the ten minute mean values of power and calibrated wind speed. The wind speed was normalized with a value between 10 and 14 m/s for confidentiality reasons.

The traceability of the measurements of the spinner anemometer on T4 was ensured by the calibrated met-mast instruments and the NTF, while the traceability of the spinner anemometer on T5 was ensured by the NTF and wind tunnel calibration of the sonic sensors.

The air density was calculated from the met-mast measurements with Eq. 4 (from IEC61400-12-2 (2013)), where $P_w = 0.0000205 \cdot e^{(0.06138467 \cdot T)}$, $R_0 = 287.05$ J/kg K, and $R_w = 461.5$ J/kg K. $T$ expressed in Kelvin, $P$ in absolute Pascal.

$$\rho = \frac{1}{T}\left(\frac{P}{R_0} - RH \cdot P_w \left(\frac{1}{R_0} - \frac{1}{R_w}\right)\right) \tag{4}$$

Measured air density was between 1.2 and 1.27 kg/m$^3$.



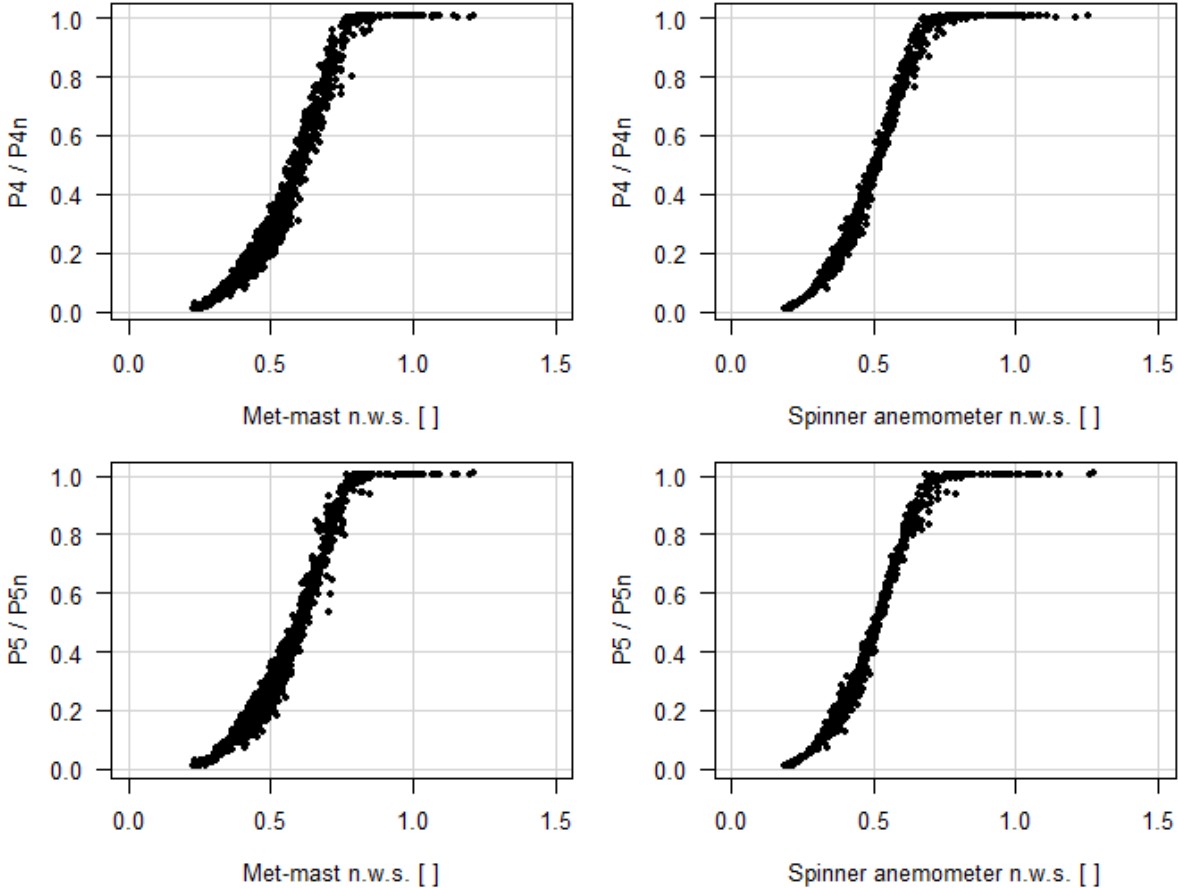

**Figure 5.** Scatter plot of power as a function of spinner anemometer normalized wind speed (n.w.s.) and met-mast normalized wind speed measurements, for turbines T4 and T5, before application of the NTF and before air density correction. Met-mast measurements to the left, and spinner anemometer measurements to the right. Turbine T4 measurements upper and turbine T5 measurements lower. Data refers to the same measurement period.

## 5   Nacelle transfer function measurement

The purpose of the NTF is to correct the spinner anemometer measurements to be representative of the free wind speed. $U_{mm}$ is the free wind speed measured by the met-mast, and $U_{free}$ is the free wind speed calculated by correcting the spinner anemometer measurements ($U_{hor}$) with the NTF.

5   The IEC61400-12-2 (2013) standard defines the NTF as the met-mast wind speed binned as a function of the nacelle wind speed. Krishna et al. (2014) investigated the root cause for high deviations in the self consistency check with the IEC61400-12-2 (2013) method and proposed an improved method, which consist of binning the spinner anemometer wind speed as a function of the met-mast wind speed. This procedure is used here. If a wind speed bin has less than 3 measurements, the value





of the NTF is calculated by linear interpolation from the adjacent bins if they both have at least 3 measurements each. No air density correction was made for the measurement of the NTF. The measured NTF for the spinner anemometer installed on turbine T4 is shown in Fig. 6 .

As expected the NTF is approximately 1:1 at high wind speed (around 11-15 m/s, thanks to the $k_1$ calibration), and is lower than 1:1 for the range of wind speeds where the turbine is operating with high Cp (high induction, which makes the wind speed by the spinner anemometer lower than the free wind speed).

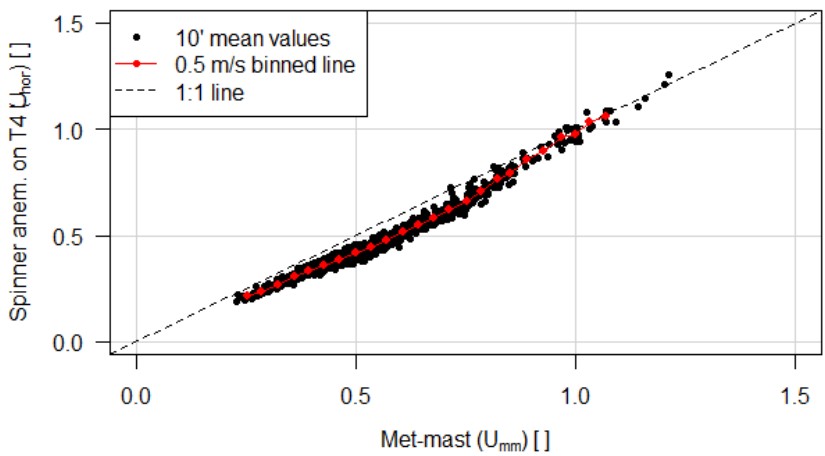

**Figure 6.** Nacelle transfer function measured with the spinner anemometer of turbine 4. Red line is the NTF obtained by linear interpolation between the red dots, which are the NTF binned values.

## 6  NTF self consistency check

The black line in Fig. 7 shows the power difference between the PC and the NPC of turbine 4. The blue and red curves shows the pass/fail boundaries defined in IEC61400-12-2 (2013) for the NTF. Both power curves were interpolated to the center of the bin with a cubic spline[1], so that the power values for the two power curves correspond to the same wind speed. Krishna et al. (2014) claimed that a NPC calculated from the same data-set used to measure the NTF (as it is the case for turbine 4) is identical to the PC (and therefore the self consistency check should return zero power difference for any wind speed bin). However in the present calculations the power difference was not zero. The PC was binned according to the met-mast wind speed ($U_{mm}$), and the NPC was binned according to the corrected nacelle wind speed ($U_{free}$). Krishna et al. (2014) suggested to bin both PC and NPC according to $U_{mm}$ to keep uniformity in the binning process, but doing so would mean binning the exact same measurements for NPC and PC, resulting obviously in the same binned values of power.

---

[1]as suggested in the draft of IEC61400-12-1, 88/460/CD, regarding presenting a power curve with values interpolated to the center of the bin.




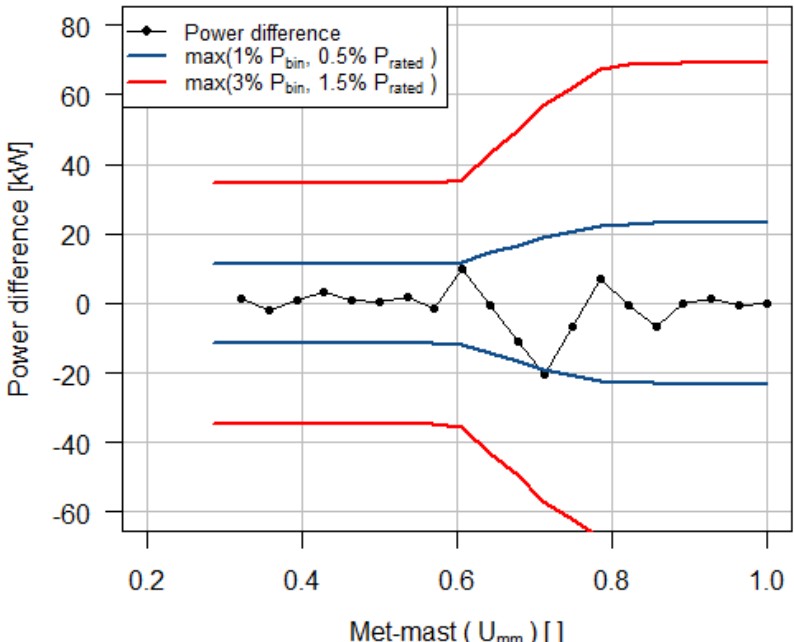

**Figure 7.** Nacelle transfer function self consistency check. The black curve shows the power difference between NPC and PC of the wind turbine used to measure the NTF. The NTF passes the test if the black curve is within the boundary marked by the blue curve. If the black curve crosses the red curve a new NTF shall be measured.

As mentioned, Krishna et al. (2014) suggest to bin the NTF corrected nacelle wind speed according to the met-mast wind speed $U_{mm}$ to check the validity of the NTF. In the normal use of the NTF the met-mast is not available, and the power curve would be binned according to $U_{free}$. The procedure to calculate a NPC shall be the same on the reference turbine (where the NTF was measured and verified with the self consistency check) and on other turbines. Therefore it makes make more sense always to calculate the bin averaged power curve binning according to $U_{free}$. In the procedure used in the present analysis, the bin average of the NTF corrected nacelle wind speed $U_{free}$ are different from the bin average of the measured free wind speed $U_{mm}$ (binning both according to $U_{mm}$). The cause is explained as follows.

The bin averages are computed by binning according to the same $U_{mm}$, therefore the binning itself should not make a difference. The spinner anemometer measurements that fall outside the range of definition of the NTF are lost during the application of the NTF. Therefore the bin average of those utmost bins will most likely be different from the original bin average value. One more reason for the bin average values to be different is that the correction applied with the NTF is applied to the time series trough a linear interpolation, not to the bin average value. The binned values of $U_{free}$ and $U_{mm}$ would be equal only if the NTF correction was constant for all the measurements of the bin with a value corresponding to the NTF. When



the NTF is applied to the time series, the slopes of the linear interpolation segments are different on the two sides of the NTF definition point in a certain bin $i$. In fact in Fig. 9 top-right the red line is not a horizontal straight line.

## 7 Application of the nacelle transfer function

The NTF measured on turbine 4 was applied on spinner anemometer measurements of turbine 4 and then on turbine 5. Linear
interpolation was used between the points that defines the NTF as described in the IEC61400-12-2 (2013) standard. The measurements that fall outside the range of the definition of the NTF are lost, since the NTF is undefined for these measurements. With the application of the NTF, part of the measurements were lost because the NTF was not defined above a certain wind speed (in Fig. 6 the red line does not extend as much as the black points, therefore about 2.5 hours of measurements are lost out of 237 hours).

The relation between free wind speed measured from the met-mast $U_{mm}$ and free wind speed calculated from spinner anemometer measurements $U_{free}$ is shown in the scatter plot of Fig. 8 for turbines 4 and 5.

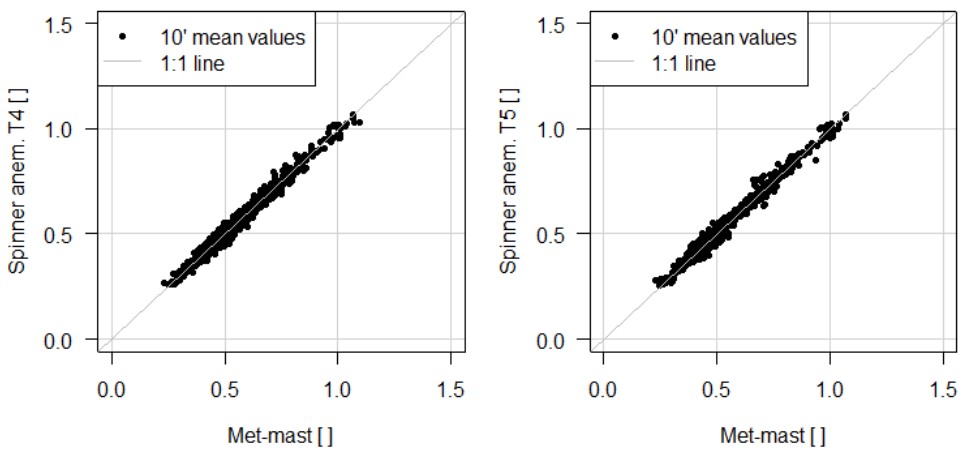

**Figure 8.** Calculated free wind speed as a function of measured free wind speed. Turbine 4 to the left and turbine 5 to the right. $R^2_{T4} = 0.9839579$, $R^2_{T5} = 0.9845664$.

Since the spinner anemometer was calibrated following the method described in section 3, the spinner anemometer wind speed measurements are already matching the met-mast wind speed at high wind speeds (U > 1.2 times rated wind speed), that is when the rotor induction is low. From Fig. 9 we can see that the correction applied by the NTF is mostly localized below
rated wind speed.



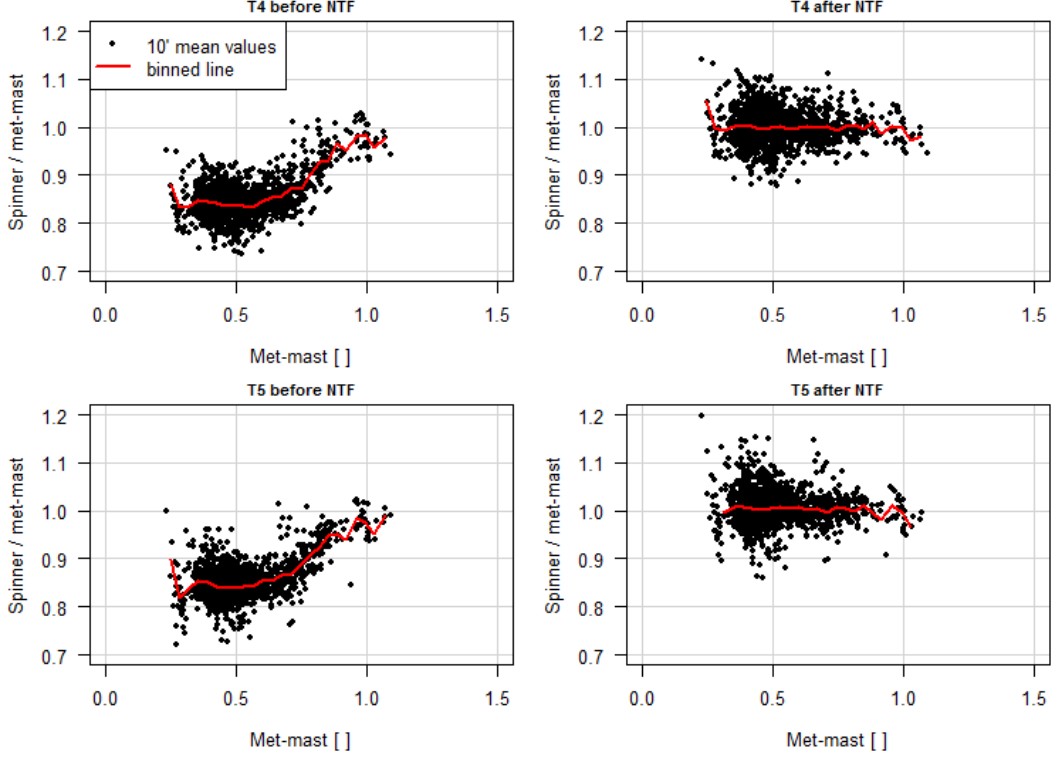

**Figure 9.** Ratio between calculated free wind speed and measured free wind speed ($U_{free}/U_{mm}$) as a function of measured free wind speed ($U_{mm}$).

## 8 Power curves and AEP

The calculated free wind speed and measured free wind speed were corrected to standard air density of 1.225 kg/m$^3$ with Eq. 5 after the application of the NTF. This is in accordance with IEC61400-12-2 (2013) for a pitch regulated turbine.

$$U_{free,n} = U_{free} \left( \frac{\rho}{1.225} \right)^{1/3} \tag{5}$$

5   The met-mast power curve was also corrected to standard air density of 1.225 kg/m$^3$ with Eq. 6 in accordance with IEC61400-12-1 (2005) for a pitch regulated turbine.

$$U_{mm,n} = U_{mm} \left( \frac{\rho}{1.225} \right)^{1/3} \tag{6}$$

The power curves of turbines 4 and 5 were obtained by averaging the power in each wind speed bin of 0.5 m/s, see Fig. 10. The value of power was interpolated with a cubic spline to the center of the wind speed bin so that the power values of the four

10   power curves are comparable (they all refer to the center of the wind speed bins).

Figure 10 shows the four power curves, NPC for T4, PC for T4, NPC for T5, PC for T5.





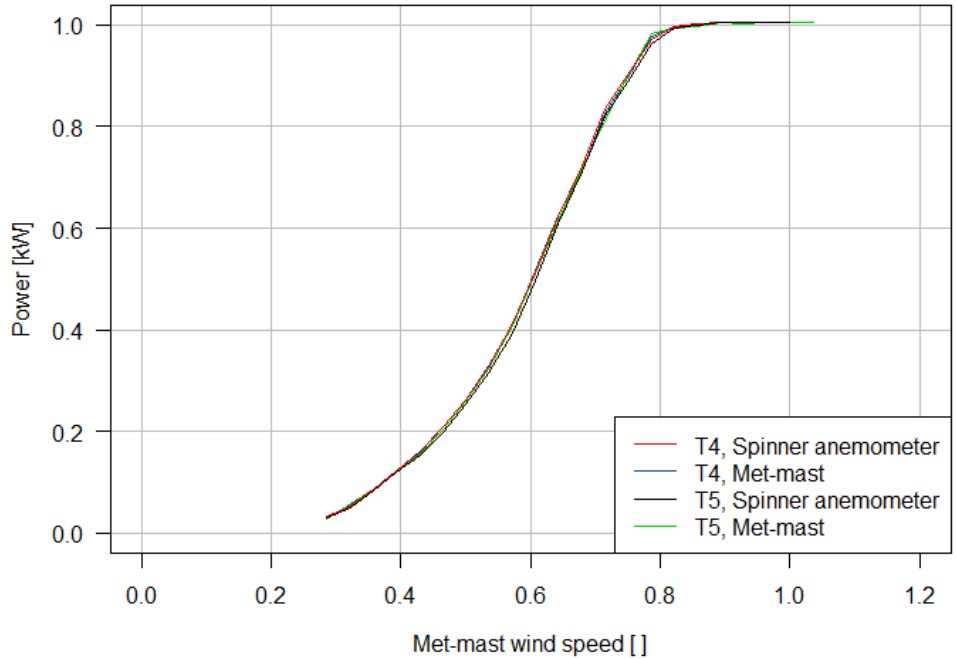

**Figure 10.** Power curves of turbines 4 and 5, measured with met-mast and with spinner anemometer.

A measure of the difference between the curves was evaluated by calculating the annual energy production (AEP) for a Rayleigh wind speed distribution with annual average wind speed between 4 m/s and 11 m/s. Table 2 shows the difference in AEP estimated for the four power curves. The AEP was calculated for the extrapolated power curve up to 25 m/s.

The nacelle power curve compared with the met-mast power curve within 0.10 % of AEP (at 8 m/s average wind speed) on the reference turbine, and within 0.38 % on the other turbine (see Tab. 2). The NPC is not identical to the PC, as well as the binned values of $U_{free}$ are not equal to the binned values of $U_{mm}$ even when binning both according to $U_{mm}$.

As expected, PC4 with NPC4 compares better than PC5 with NPC5, since the NTF was measured on T4. The uncertainty on AEP calculated for PC4 in the DTU report I-0440 Demurtas (2015) (with the same turbine and same measurement set-up, but not public for confidentiality of the data presented) was found to be 14.2% for $V_{avg} = 4$ m/s, 5.7% for $V_{avg} = 8$ m/s and 4.2% for $V_{avg} = 11$ m/s. The AEP difference is more than ten times smaller than the AEP uncertainty.



**Table 2.** Comparison between met-mast power curve (PC) and nacelle (spinner) power curve (NPC) in terms of annual energy production. The values in the table are calculated as $(from/to - 1) \cdot 100$. The AEP was calculated with the extrapolated power curve from valid data to 25 m/s.

| $V_{avg}$ | From: | NPC4 | NPC5 | PC5 | NPC5 |
| --- | --- | --- | --- | --- | --- |
| | to: | PC4 | PC5 | PC4 | NPC4 |
| m/s | | % | % | % | % |
| 4 | | 0.10 | -1.35 | -1.04 | -2.47 |
| 5 | | 0.14 | -0.95 | -0.74 | -1.82 |
| 6 | | 0.13 | -0.69 | -0.55 | -1.37 |
| 7 | | 0.12 | -0.51 | -0.44 | -1.06 |
| 8 | | 0.10 | -0.38 | -0.36 | -0.84 |
| 9 | | 0.09 | -0.30 | -0.31 | -0.69 |
| 10 | | 0.07 | -0.24 | -0.28 | -0.59 |
| 11 | | 0.07 | -0.19 | -0.25 | -0.51 |

## 9  Uncertainty analysis

This section will describe the evaluation of uncertainty of the free wind speed measured with the met-mast, and free wind speed calculated with an NTF applied to spinner anemometer measurements. The spinner anemometer measures the wind speed by means of three sonic sensors and a conversion algorithm. Each sensor was calibrated independently in a MEASNET compliant wind tunnel. The uncertainty of the three velocities were combined through the spinner anemometer conversion algorithm to give the uncertainty of the horizontal wind speed.

The uncertainty of spinner anemometer on T4 due to differences in mounting of the three sonic sensors is zero, since this spinner anemometer was used to measure the NTF. The mounting of the second spinner anemometer (on T5) was compared with the mounting of the reference spinner anemometer (on T4) and an additional uncertainty due to mounting differences with respect to T4 was added to the measurements of the spinner anemometer on T5.

### 9.1  Uncertainty related to wind tunnel calibration of sonic sensors

The relation between the wind tunnel speed and the velocity component in the sensor path is:

$$V_1 = V_t \cdot \cos(\phi_s) \tag{7}$$

If the angle $\phi_s$ of the sonic sensor path with respect to the horizontal mounting plate was not measured, one should assume that $\phi_s$ is within the manufacturing tolerance, $\phi_s = 35° \pm 1.5°$. The standard uncertainty on $\phi_s$ can therefore be expressed by the tolerance divided by the square root of three as:

$$u_{\phi s} = (a_+ - a_-)/(2\sqrt{3}) = (36.5 - 33.5)/(2\sqrt{3}) = 0.866° \tag{8}$$



In this case the angle $\phi_s$ was measured as part of the wind tunnel calibration (see Tab. 1). The uncertainty on $\phi_s$ depends on the accuracy of the protractor (the instrument to measure angles). In this case, a digital protractor with an accuracy of $0.2°$ was used, and therefore $u_{\phi s} = 0.2°$ was used instead of $0.866°$ .

The uncertainty on the wind tunnel calibration was expressed in the calibration certificates for a coverage factor $k_c = 2$ as a binned value as a function of wind tunnel speed. While the uncertainty is typically almost constant for a cup anemometer, the sonic sensor uncertainty showed increase with wind speed. The standard uncertainty ($k_c = 1$) was calculated by dividing the value reported in the certificates by two. The calibration standard uncertainty (function of wind speed) was fitted to a line as shown in Eq. 9.

$$u_t = (2.24 \cdot V_i + 0.855) \cdot 10^{-3} m/s \tag{9}$$

The uncertainty on the sonic sensor velocity $V_1$ is obtained combining the uncertainty $u_t$ with the uncertainty $u_{\phi s}$, using the equation for combination of uncertainty of independent variables as expressed in Eq. 10 (according to section 5.1.2 of the GUM, JCGM/WG1 (2008)) and also shown in IECRE (2015) clarification sheet.

$$u_c^2(y) = \sum_{i=1}^{N} \left( \frac{\partial f}{\partial x_i} \right) u^2(x_i) \tag{10}$$

Equation 10 applied to Eq. 7 results in Eq. 11:

$$u_1^2 = \left( \frac{\partial (V_t \cos \phi_s)}{\partial V_t} \right)^2 u_t^2 + \left( \frac{\partial (V_t \cos \phi_s)}{\partial \phi_s} \right)^2 u_{\phi s}^2 = \cos^2 \phi_s \cdot u_t^2 + V_t^2 \sin^2 \phi_s \cdot u_{\phi s}^2 \tag{11}$$

With $u_t$ (Eq. 9) as the uncertainty of the wind tunnel wind speed and $u_{\phi s}$ as the uncertainty on the sensor path angle (Eq. 8 or uncertainty of the protractor). The combined uncertainty on $V_1$ due to wind tunnel calibration is:

$$u_1 = \sqrt{(\cos \phi_s)^2 \cdot u_t^2 + (V_t \cdot \sin \phi_s)^2 \cdot u_{\phi s}^2} \tag{12}$$

The same applies to each of the sensors ($u_2$, $u_3$).

**9.2  Evaluation of spinner anemometer mounting**

The three sonic sensors should be mounted on the spinner with the best possible rotational symmetry and equal distance from the spinner center of rotation. A visualization method for documentation of the sonic sensors installations was developed by Demurtas and Pedersen with the use of photography, Demurtas (2015). The initial mounting of the sensors was used for the first power curve measurements reported in Demurtas (2015). The accuracy of sensor mounting was then improved and the 25 power curve measurement repeated and reported in this work. The mounting position was evaluated with photography method described in . Due to the challenge of photographing a feature of size in the order of centimeters (the sonic sensor) from a long distance (80 meters from ground to spinner) we used a 400 mm optic zoom and a high resolution digital camera (24 megapixel).





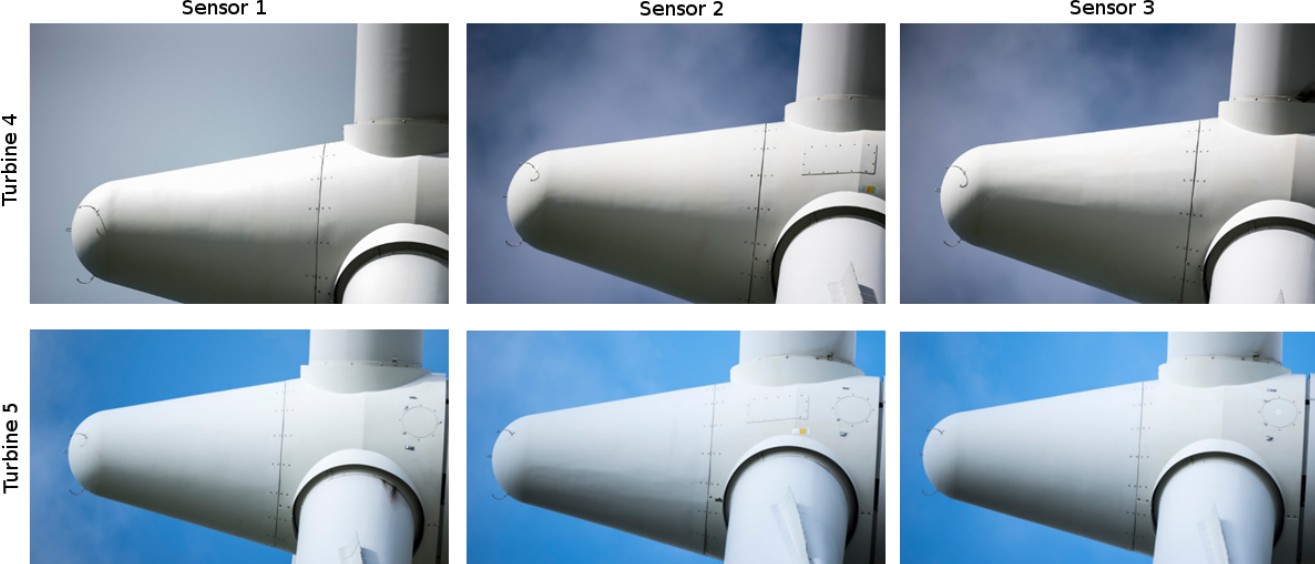

**Figure 11.** Photos of two spinner anemometers (turbine 4 above, turbine 5 below. Sensors 1, 2 and 3 from left to right.).

Several photos of the spinner were taken from the ground during rotation of the wind turbine and three photos selected each time a sonic sensor is visible on the side of the spinner, with the sky in the background.

Each of the six photos (three for each turbine, Fig. 11) was post processed making it semi-transparent. The photos were overlayed, scaled and rotated in order to make the spinner contour to match. The sky was made transparent and a contrasting

5   red background added (Fig. 12).

The photo overlay was scaled to make the sonic sensor path 16.7 cm long, like it is in reality. The positions of the sonic sensors on the spinner were measured in the plane of the photos as the angle between a plane perpendicular to the spinner axis and the sensors of extreme forward and backwards positions. The position of the sensor paths were measured on the photo with a vector graphic software (inkscape).

10   The improved mounting of the sensors showed a mounting accuracy in the order of $\pm 2$ cm. This was an improvement of the initial mounting whose accuracy was $\pm 6$ cm. The sensors of the two turbines fell into a mounting angle interval $[a_-, a_+] = [31°, 40°]$ for the old mount, and $[48°, 51°]$ for the improved new mount. In the improved mount the sensors were also moved forward on the spinner, for practical reasons, not to interfere with the old mounting holes.





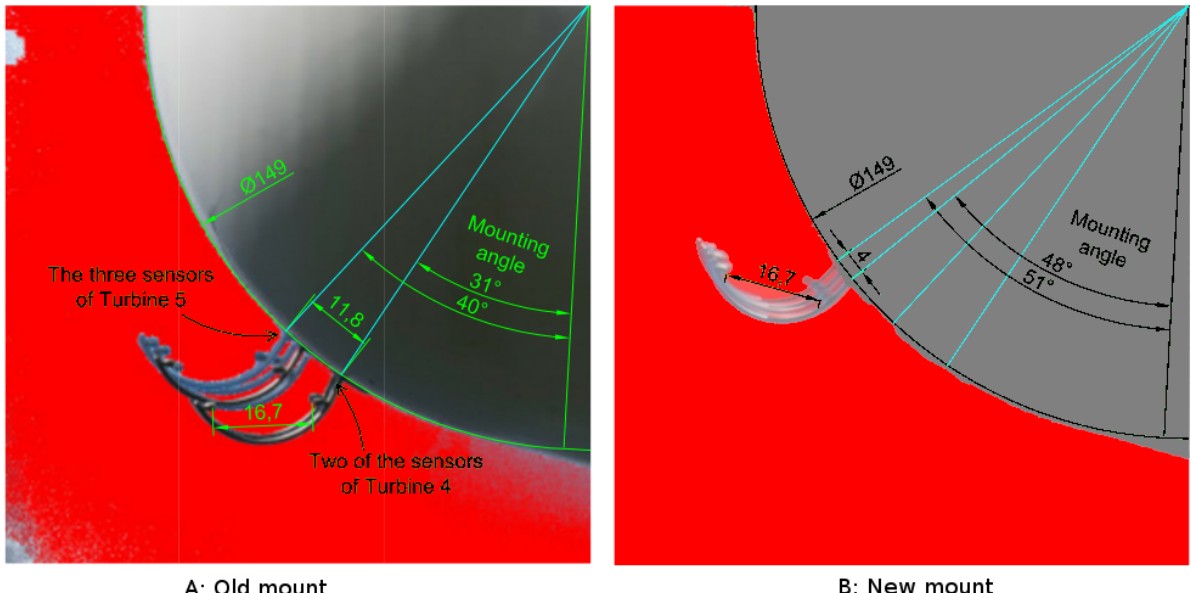

**Figure 12.** Sonic sensors relative mounting position between turbine 4 and turbine 5. Not to scale, dimensions are in centimeters. Left: original mounting. Right: after improvement of the mounting accuracy. The blue lines connects the center of the spinner sphere with the extreme mounting position of the sensors. In the figure at the right hand side four blue lines show the position of the original and improved mounting.

### 9.3 Uncertainty in wind speed measurements due to mounting imperfections

The uncertainty connected to the error in mounting position of the sonic sensor was investigated approximating the spinner as a sphere and using potential flow theory to calculate the flow around a sphere. The mean air velocity along the sensor path was calculated averaging the wind velocity component along the sensor path in three points along the path (points shown with a black or red dots in Fig. 13).

The flow field was calculated for a mesh of 0.01 in $x$ and $y$ direction. The coordinates of each point were converted from cartesian coordinates $(x_p, y_p)$ to polar coordinates $(r, \theta)$ with Eq. 13 and Eq. 14. An angle of $\pi/2$ was added to the $\theta_p$ coordinate (Eq. 14) to rotate the result in order to have the flow coming from the left parallel to the $x$ axis. This also rotated the origin of the angles to the vertical axis, which is convenient to measure the position of the mounting angles of the sonic sensors.

$$r = \sqrt{(x_p^2 + y_p^2)} \tag{13}$$

$$\theta_p = \arctan(y_p/x_p) + \pi/2 \tag{14}$$





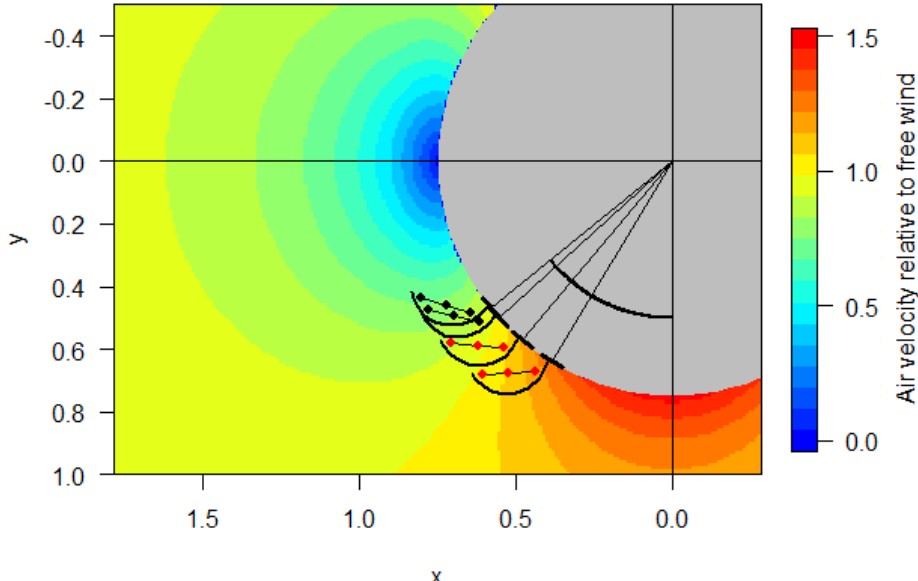

**Figure 13.** Simulation of the flow around the spinner, which was approximated to a sphere. The flow was calculated with the equations of the potential flow around a sphere. The sensor positions retrace the position of the sensors found in Fig. 12. The sensor paths with the red dots refer to the old mount $(31°, 40°)$, while these with the black dots refer to the improved mounting $(48°, 51°)$, where the sensors are more closely spaced.

The potential flow model is oriented such that the inflow is in axis with the pinner axis of rotation, therefore $U = U_{hor} = U_0$. The flow field was calculated in polar coordinates with Eq. 15 (along radius) and Eq. 16 (perpendicular to radius) with the equations by Faith and Morrison (2013).

$$v_r = U_0[1 - (R/r)^3]\cos(\theta_p) \tag{15}$$

$$v_t = -U_0[1 + 0.5 \cdot (R/r)^3]\sin(\theta_p) \tag{16}$$

The modulus of the wind speed was calculated with Eq. 17 and shown with the color scale in Fig. 13.

$$U = \sqrt{(v_r^2 + v_t^2)} \tag{17}$$

The air velocity along the sensor path at the point $p(r,\theta)$ was calculated with Eq. 18, where $35°$ is the default angle between the sensor path and the sensor root (tangent to the spinner surface).

$$U_p = v_r \cdot \sin(35 \cdot \pi/180) - v_t \cdot \cos(35 \cdot \pi/180) \tag{18}$$





Equation 18 was used to calculate the wind velocity along the sensor path in each of the black points shown in Fig. 13. Then, the mean wind speed along each sensor path was calculated as average value of the velocity along the sensor path in the three points related to each sensor path (Eq. 19).

$$U_{path} = \frac{U_{p1} + U_{p2} + U_{p3}}{3} \tag{19}$$

The sensor path wind speed was calculated for each of the four sensor mounting positions measured with the photographic method of Fig. 12, and is presented in Tab. 3. The sensor path wind speed was normalized to the wind speed upstream of the spinner ($U_0$). The uncertainty due to mounting imperfections is a type B uncertainty ("Guide to the expression of uncertainty in measurements" (JCGM/WG1 (2008), chapters 4.3.1 and 4.3.7). The probability of the sonic sensor wind velocity to be within the interval $a_-$ to $a_+$ calculated from the positions identified with the photographic method is equal to one and the probability

that it lies outside the interval is zero. There is no particular reason for the sensor path wind velocity to fall into the interval in a particular position. Therefore we can assume that the probability that the sensor path wind speed is within the interval is a rectangular distribution. This means that the standard uncertainty is:

$$u_m = (a_+ - a_-)/(2\sqrt{3}) \tag{20}$$

The uncertainty is a value relative to the wind speed upstream of the spinner anemometer ($U_{hor}$). This uncertainty does not

apply to the wind turbine 4 which was used to measure the NTF.

**Table 3.** Mounting angles, sensor path wind speed and uncertainty due to mounting accuracy.

| | Initial mount | | Improved mount | |
|---|---|---|---|---|
| Mounting angle | 31° | 40° | 48° | 51° |
| Sensor path relative speed ($U_{path}/U_0$) | $a_+ = 0.9864$ | $a_- = 0.8940$ | $a_+ = 0.7934$ | $a_- = 0.7516$ |
| Uncertainty of sensor path wind speed ($u_m$) | 2.7% $U_{hor}$ | | 1.2% $U_{hor}$ | |

### 9.4    Combination of uncertainties through the spinner anemometer conversion algorithm

This section will explain how to combine the uncertainty on the input quantities to obtain the uncertainty on the output of the spinner anemometer: the horizontal wind speed. The uncertainty on $U_{hor}$ is the combination of the following uncertainty components:

- $u_1$ Sensor 1 wind tunnel calibration (which includes $u_t$ and $u_{\phi s}$).

- $u_2$ Sensor 2 wind tunnel calibration (which includes $u_t$ and $u_{\phi s}$).

- $u_3$ Sensor 3 wind tunnel calibration (which includes $u_t$ and $u_{\phi s}$).

- $u_m$ Sensors mounting.



   – $u_{k\alpha}$ Angular calibration.

The uncertainty on $k_1$ is $u_{k1} = 0$ because all the uncertainty related to wind speed is included in the uncertainty of the NTF ($u_{NTF}$, see section 9.7). The uncertainties on the sonic sensor speeds ($u_1$, $u_2$ and $u_3$) are substantially equal but we keep them separated with different names for clarity. $U$ is not measured directly, but is determined from the quantities $V_{ave}$ and $\alpha$ through a functional relationship $g$:

$$U = g(V_{ave}, \alpha) = \frac{V_{ave}}{k_1 \cos \alpha} \tag{21}$$

$\alpha$ is also not measured directly but is determined from the quantities $V_1$, $V_2$, $V_3$ and $k_\alpha$ through a functional relationship $f$:

$$
\begin{aligned}
\alpha = f(V_1, V_2, V_3, k_\alpha) &= \arctan\left(\frac{k_1 \sqrt{3(V_1 - V_{ave})^2 + (V_2 - V_3)^2}}{\sqrt{3} k_2 V_{ave}}\right) = \\
&= \arctan\left(\frac{2}{k_\alpha} \frac{\sqrt{(V_1^2 + V_2^2 + V_3^2 - V_1 V_2 - V_1 V_3 - V_2 V_3)}}{V_1 + V_2 + V_3}\right)
\end{aligned}
\tag{22}
$$

$V_{ave}$ is the average between $V_1$, $V_2$, $V_3$ calculated with the relationship $h$:

$$V_{ave} = h(V_1, V_2, V_3) = \frac{1}{3}(V_1 + V_2 + V_3) \tag{23}$$

To calculate the uncertainty on $U$ we need first to calculate the uncertainty on $V_{ave}$ and on $\alpha$.

The uncertainty on $V_{ave}$ is calculated applying the rule for combination of uncertainties of uncorrelated input quantities (Eq. 10) to the function $h$ (Eq. 23), resulting in Eq. 24 assuming that $u_1 = u_2 = u_3$.

$$u_{ave} = \sqrt{\left(\frac{1}{3}\right)^2 u_1^2 + \left(\frac{1}{3}\right)^2 u_2^2 + \left(\frac{1}{3}\right)^2 u_3^2} = \frac{u_1}{\sqrt{3}}. \tag{24}$$

The uncertainty on the inflow angle $\alpha$ can be calculated combining the uncertainty of $V_1$, $V_2$, $V_3$, and $k_\alpha$ applying Eq. 10 to the function $f$ (Eq. 22), resulting in Eq. 25.

$$u_\alpha = \sqrt{\frac{\partial f}{\partial V_1} u_1^2 + \frac{\partial f}{\partial V_2} u_2^2 + \frac{\partial f}{\partial V_3} u_3^2 + \frac{\partial f}{\partial k_\alpha} u_{k\alpha}^2} = \sqrt{3 \frac{\partial f}{\partial V_1} u_1^2 + \frac{\partial f}{\partial k_\alpha} u_{k\alpha}^2} \tag{25}$$

Given the complexity of the function $f$ (Eq. 22) the derivative was computed numerically with the help of a computer.

$V_1$, $V_2$, $V_3$ were calculated for a wind speed $U$ in a range 0-25 m/s with Eq. 27 to 29, for six arbitrary values of $\alpha$, and used to compute the partial derivatives of Eq. 25. The uncertainty on $k_a$ was set to $u_{k\alpha} = 0.1 \cdot k_\alpha$ as found by Pedersen et al. (2015). In Fig. 14 one can see an uncertainty of about 1° for a inflow angle of 10°.





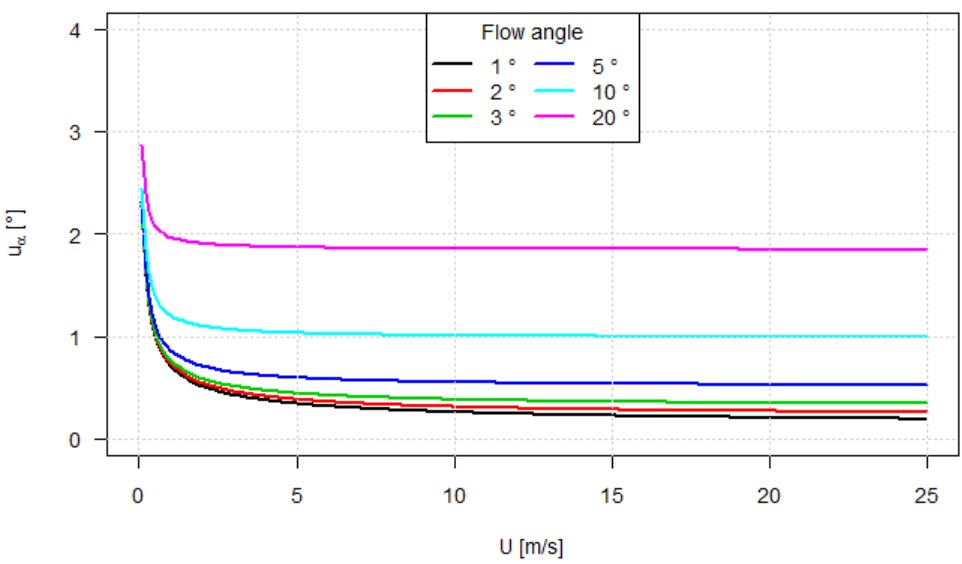

**Figure 14.** Uncertainty on the inflow angle $\alpha$.

The uncertainty of the vector wind speed $U$ can be calculated applying the method for combination of uncertainty of independent variables (GUM JCGM/WG1 (2008), Eq. 10) to the function $U = g(V_{ave}, \alpha)$ (Eq. 21), resulting in Eq. 26.

$$u_U = \sqrt{\left(\frac{\partial g}{\partial V_{ave}}\right)^2 u_{ave}^2 + \left(\frac{\partial g}{\partial \alpha}\right)^2 u_\alpha^2} = \sqrt{\left(\frac{1}{k_1 \cos \alpha}\right)^2 u_{ave}^2 + \left(\frac{V_{ave}}{k_1} \frac{\sin \alpha}{\sqrt{1 + \alpha^2}}\right)^2 u_\alpha^2} \tag{26}$$

The uncertainty on $U$ calculated for six arbitrary values of $\alpha$ with Eq. 26. $V_{ave}$ was calculated with Eq. 23 and $V_1$, $V_2$ and $V_3$

5   with Eq. 27, 28 and 29. The results are shown in Fig. 15.

$$V_1 = U(k_1 \cos(\alpha) - k_2 \sin(\alpha) \cos(\theta)) \tag{27}$$

$$V_2 = U\left(k_1 \cos(\alpha) - k_2 \sin(\alpha) \cos(\theta - \frac{2\pi}{3})\right) \tag{28}$$

10   $$V_3 = U\left(k_1 \cos(\alpha) - k_2 \sin(\alpha) \cos(\theta - \frac{4\pi}{3})\right) \tag{29}$$



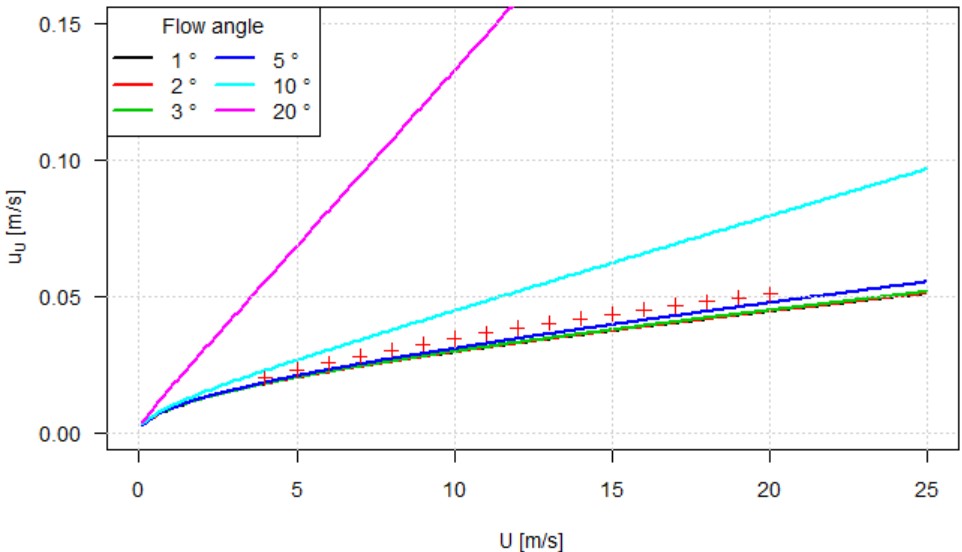

**Figure 15.** Uncertainty on wind speed ($u_U$) as a function of wind speed ($U$), for six possible values of inflow angle $\alpha$. Red crosses shows the values of uncertainty as a function of wind speed for a common value of inflow angle of $6°$.

As seen in Fig. 15, the uncertainty on $U$ is function of the flow angle $\alpha$. For inflow angles below $5°$ Fig. 15 shows that the $u_U$ is basically only function of $U$. A typical average inflow angle to a wind turbine is smaller than $5°$, as presented in Pedersen et al. (2014). The uncertainty of the wind speed is typically function of the wind speed only. In order to keep the calculation simple (especially in the calculatio of power curve uncertainty), a simple model (Eq. 30, red crosses in Fig. 15) was fitted to

5 the line corresponding to an inflow angle of $6°$, which is unlikely to be exceeded on average during normal operation of most wind turbines in a range of wind speeds 4 to 20 m/s.

$$u_U = -0.005 + \frac{\sqrt{U}}{80}. \tag{30}$$

Now that the uncertainty on the wind speed modulus $U$ is known, it is possible to calculate the uncertainty on its horizontal component $U_{hor}$. By combining the equations of the conversion algorithm (which can be found in Demurtas et al. (2016)),

$U_{hor}$ is expressed as:

$$U_{hor} = i(U, \delta, \phi, \theta, \alpha) = \sqrt{(U \cos\alpha \cos\delta - U \sin\alpha \sin(\phi + \theta)\sin\delta)^2 + (-U \sin\alpha \sin(\phi + \theta))^2} \tag{31}$$

The position of the flow stagnation point $\theta$ in Eq. 31 is a function of $V_1$, $V_2$, $V_3$. The rotor position $\phi$ is calculated based on the accelerometers located in each sonic sensor root. To be absolutely correct, one should apply the method for combination of uncertainty to Eq. 31. However, it is reasonable to assume that $U_{hor} \sim U$ due to the small inflow angle $\alpha$ and that $u_{Uhor} \sim u_U$

because the uncertainty on the turbine tilt angle $\delta$ and rotor position $\phi$ is likely to be smaller than the other uncertainty





components. Moreover, the improved accuracy in the estimation of $u_{Uhor}$ would be wiped out by the simplification made with Eq. 30. Therefore, the uncertainty on the horizontal wind speed $U_{hor}$ is reasonably equal to the uncertainty on $U$:

$$u_{Uhor} = u_U \tag{32}$$

### 9.5 Uncertainty of spinner anemometer output

5  The uncertainty of the spinner anemometer wind speed measurements of turbine 4 (Eq. 33) is the combination of the uncertainty on the spinner anemometer output ($u_{Uhor}$) with the uncertainty due to the discrepancies between different MEASNET wind tunnels ($U_{ME} = 1\%/\sqrt{3}$).

$$u_{s.a.4} = \sqrt{u_U^2 + u_{ME}^2} \tag{33}$$

The uncertainty on the measurements of the second spinner anemometer (on T5) shall also include the uncertainty due to

10  mounting imperfections to account for the dissimilarity with the reference spinner anemometer (on T4):

$$u_{s.a.5} = \sqrt{u_U^2 + u_{ME}^2 + u_m^2} \tag{34}$$

Figure 16 shows the combination of each uncertainty term to the final uncertainty budgets. As it can be seen in Fig. 16, the uncertainty of the sensor path speed (pink crosses) is very close to the wind tunnel speed (black line), due to the small contribution to the uncertainty coming from the uncertainty of the sensor path angle $\phi_s$.

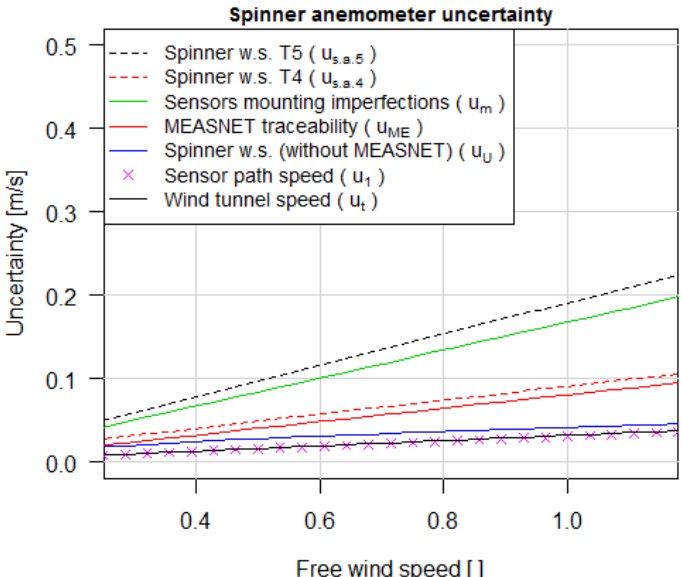

**Figure 16.** Overview of the size of the various uncertainty components and total uncertainty.





The spinner anemometer conversion algorithm combines the uncertainties of the spinner anemometer input quantities ($V_1$, $V_2$, $V_3$, $\phi$) resulting in the blue line of Fig. 16. Once combined with the MEASNET traceability uncertainty (red line) we arrive to the dashed red line. Once including the uncertainty of the mounting imperfection (green line) we arrive at the black

dashed line. Among the uncertainty components ascribable to the spinner anemometer, the one due to mounting inaccuracy of the sensors is the largest one. Note that the mounting imperfections are null for the reference spinner anemometer, in fact what matters is that the mounting position of the sonic sensors and the shape of the other spinner anemometers (on T5 in this case) are similar to the reference one (on T4 in this case). All the sonic sensors were calibrated in the same wind tunnel. The MEASNET uncertainty was added to $u_U$ instead of to $u_1$, $u_2$ and $u_3$ to avoid counting it three times.

## 9.6    Uncertainty of met-mast measurements

The uncertainty of the met-mast wind speed measurement (Eq. 35, Fig. 17) is (according to IEC61400-12-2 (2013)) the combination of the wind tunnel uncertainty $u_t$, the MEASNET uncertainty to account the discrepancies between different wind tunnels $u_{ME}$, the uncertainty due to the cup anemometer class $u_{a.class}$ (that takes into account the response of the cup anemometer to turbulence and flow inclination), and $u_{s.cal.} = 2\% V_i$ because there was no site calibration.

$$u_{mm} = \sqrt{u_t^2 + u_{ME}^2 + u_{a.class}^2 + u_{s.cal.}^2} \tag{35}$$

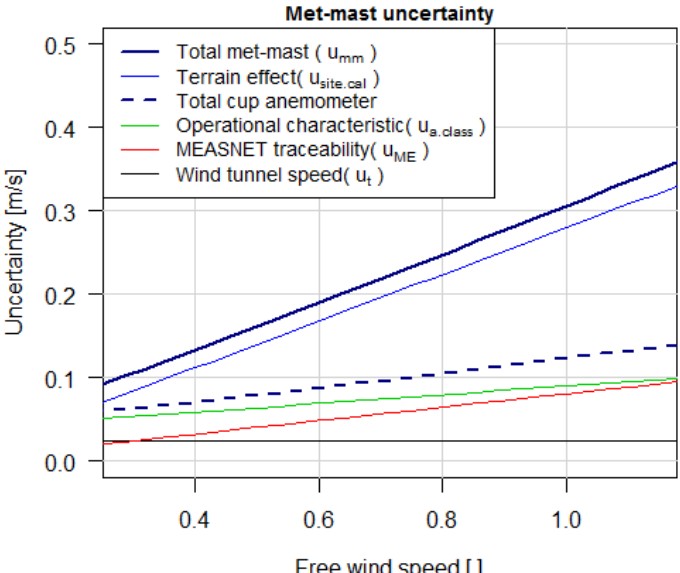

**Figure 17.** Overview of the size of the various uncertainty components and total uncertainty.



## 9.7 Uncertainty of NTF

The uncertainty on the NTF ($u_{NTF}$) is the combination of the various uncertainty components as:

$$u_{NTF} = \sqrt{u_{mm}^2 + u_{sa4}^2 + u_M^2 + s_{NTF}^2} \tag{36}$$

where

$u_{mm}$  is the uncertainty on the measured free wind speed.

$u_{sa}$  is the uncertainty of the spinner anemometer measurements.

$u_M$  is the uncertainty due to the NTF method, considered 2% of the wind speed due to seasonal variations ($u_M = 0.02V_i$) in the standard IEC61400-12-2.

$s_{NTF}$  is the statistical uncertainty of the captured data-set ($s_{NTF} = \frac{\sigma_{NTF}}{\sqrt{N_j}} U_{sa4}$). $\sigma_{NTF}$ is dimensionless because it is the standard deviation of the ratio $U_{free}/U_{sa4}$.

## 9.8 Uncertainty on calculated free wind speed

To measure the absolute power curve of a wind turbine the spinner anemometer output must be corrected to free wind speed by use of the nacelle transfer function (NTF). The uncertainty on the free wind speed is therefore a combination of $u_{s.a.}$ with $u_{NTF}$.

$$u_{free5} = \sqrt{u_{sa5}^2 + u_{NTF}^2} \tag{37}$$

For the case of the reference wind turbine (T4, used to measure the NTF) the uncertainty is calculated differently. $u_{NTF}$ already contains the uncertainty of the reference spinner anemometer (T4) and the uncertainty of the met-mast measurements. Therefore the uncertainty of the free wind speed calculated with the NTF is just the uncertainty of the NTF (Eq. 38).

$$u_{free4} = u_{NTF} \tag{38}$$



## 10   Results of uncertainty analysis

The mounting accuracy was investigated by overlaying six photos of the spinner taken from ground level during rotation, each showing the corresponding sensor when it is at the side of the spinner. The photos unveil deviations in the order of $\pm 2$ cm between some of the sensors. It was expected that the mounting imperfections played a major role in the total uncertainty of

5   the second spinner anemometer. However, the contribution of other uncertainty sources combined (the $1/\sqrt{3}\%$ MEASNET traceability of wind tunnel calibrations for cup-anemometer on the met-mast and spinner anemometers sensors, the 2% for lack of site calibration) was much larger than the uncertainty due to the mounting of the sensors (which was 1.2%).

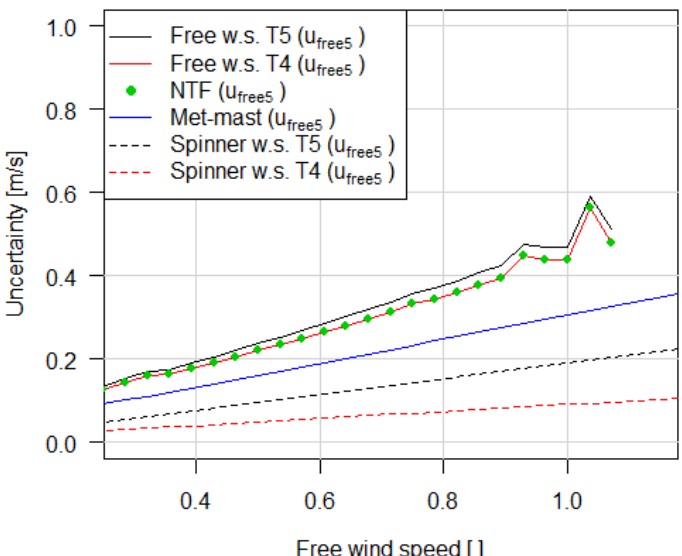

**Figure 18.** Uncertainty on wind speed. The met-mast wind speed includes 2% additional uncertainty due to lack of site calibration.

As shown in Fig. 18, the uncertainty of the NTF is larger than the met-mast uncertainty, as expected. The met-mast uncertainty is larger than the spinner anemometer uncertainties (dashed lines) because of the 2% added due to missing site

10   calibration, which does not apply to the spinner anemometer output (but applies to the NTF later to calculate the free wind speed). The turbine T5 has a larger uncertainty than the reference turbine, as expected, due to the mounting imperfections. Note that the distance between the two dashed lines is due to the mounting imperfections, but the impact of such imperfections is significantly reduced once the uncertainties on the spinner anemometer output are combined with the NTF uncertainty.





## 11 Discussion

The main goal of this study was to measure the power performance of a wind turbine using a spinner anemometer which was calibrated with the calibration determined on a "reference" spinner anemometer on an identical wind turbine. The calibration ($k_\alpha$ and $k_1$ values) determined on the reference spinner anemometer can be moved to a second spinner anemometer estimating
an additional uncertainty due to the mounting differences. This is only possible if the two spinners have the same outer shape. The mounting differences (and associated uncertainty) could be completely avoided if the positioning of the sonic sensors was exactly equal between the reference spinner, and another spinner. This geometric perfection could be achieved with the collaboration of the manufacturer of the spinner, by integrating the sensor mounting fittings in the mould, so that all the spinners comes out identical.

The spinner anemometer was calibrated for wind speed measurement, so that it reads the wind speed correctly in a condition of zero induction (stopped rotor, or operation at high wind speed ). While this step is not essential because this correction can be included in the NTF, it is convenient to use the NTF to only correct the induction. When the spinner is calibrated for wind speed measurements, a change in the spinner anemometer configuration (for example move the sensors to another point on the spinner) can be accounted with a new $k_1$, and the NTF stays unvaried.

Apply the same NTF on another turbine is reasonable only if the wind turbine control strategy and the rotor are identical to the reference turbine. This requirement however could be removed if further research can demonstrate that the induction at the rotor centre (what matters for the spinner anemometer) is unvaried for changing rotor diameter or control strategy.

The uncertainty due to discrepancies between MEASNET wind tunnels ($u_{ME} = 1\%/\sqrt{3}$) was combined with the uncertainty of the spinner anemometer output wind speed ($u_U$), while a more correct approach would have been to include $u_{ME}$ in
the wind tunnel uncertainty $u_t$. The first approach was used to keep the analysis of propagation of uncertainties (through the spinner anemometer conversion algorithm) free from contributions of constant terms (such as $u_{ME}$), which would otherwise have masked the contribution attributable to the sole spinner anemometer conversion algorithm.

Adding the MEASNET uncertainty to the spinner anemometer output instead of to the input does not lead to a significant error in the total uncertainty, since the effect of the conversion algorithm on the uncertainties is small (as shown in Fig. 16 by
the small distance between the pink crosses and the blue line, which is basically all due to the 10% uncertainty on $k_\alpha$).

If the spinner anemometer of the reference turbine (T4) is replaced, the uncertainty of the new spinner anemometer should be added to the NTF uncertainty ($u_{free4} = \sqrt{u_{NEWsa4}^2 + u_{NTF}^2}$). If the data set used to calculate NPC on T4 is different to the one used to measure the NTF, the type A uncertainty of the new data set shall be added to the NTF uncertainty ($u_{free4} = \sqrt{u_{NTF}^2 + s_{NEWsa4}^2}$).

Each sonic sensor (three for each spinner anemometer) should be calibrated in the wind tunnel and the results of the calibration set in the spinner anemometer conversion box (the procedure is explained in Demurtas (2014)). If a sensor fails and is replaced, the new wind tunnel calibration values should be set in the conversion box. If the sensors are not calibrated, a new (more difficult) calibration of $k_1$ should be made every time a sonic sensor is replaced.



The reference spinner anemometer should be calibrated in flat terrain. The calibration of the spinner anemometer for wind speed measurements and the measurement of the NTF can, in practice, be done with any free wind speed measurement device (met-mast, nacelle lidar or ground based lidar). In complex terrain, a spinner anemometer should be assigned the calibration and NTF measured on a identical wind turbine in a flat terrain. The free wind speed calculated applying the NTF to the spinner

anemometer measurements in complex terrain will provide a free wind equivalent to the one of a flat site, with no need for site calibration.

A large flow inclination can also be expected in a wind farm environment. The spinner anemometer is well suited to measure in the wake of other turbines (turbulent flow with large flow inclination angles). However it has to be kept in mind that the spinner anemometer is a point measurement, compared to the rotor swept area. If the rotor is partially operating in the wake of

another turbine there will be a reduced power output, but the spinner anemometer measurement (which is not interested by the wake) would not be representative of the average wind condition over the swept area.

## 12   Conclusions

The study investigated the methods to evaluate the power performance of two wind turbines using spinner anemometers.

The power curves of two adjacent wind turbines (T4, T5) were measured by means of a common traceable calibrated met-

mast and spinner anemometers on each turbine. All sonic sensors were calibrated in a traceable wind tunnel. T4 was the reference turbine. The reference spinner anemometer installed on T4 was calibrated with respect to angular and wind speed measurements to take into account the shape of the spinner and the mounting position of the sensors. The spinner anemometer on T5 instead, was assigned the calibration constants of the reference spinner anemometer. Similarly, the NTF (Nacelle Transfer Function) was measured on the reference turbine T4 and applied to both turbines. The four power curves of the two turbines

(two met-mast power curves and two spinner anemometer power curves) were compared in terms of AEP (Annual Energy Production). The nacelle power curves compared very well with the met-mast power curves for a range of annual average wind speeds. The uncertainty of the spinner anemometer wind speed measurements was analyzed in detail, taking account of the propagation of the uncertainty trough the spinner anemometer conversion algorithm. Some small approximations were made.

The sonic sensor mountings were verified with photos taken from the ground and a method for estimation of uncertainty

related to mounting imperfections was proposed. The uncertainty on the free wind speed calculated with the NTF was mostly due to the uncertainty of MEASNET traceability and lack of site calibration. In less significant part the uncertainty was due to the spinner anemometer sensor calibration and mounting imperfections.

In summary, under the condition that the mounting of the sonic sensors are very similar to the reference mounting, power performance measurements with use of spinner anemometer can be made within 0.38% difference in AEP for an annual average

wind speed of 8 m/s.





## Appendix A: List of symbols

$\alpha$  Wind inflow angle relative to the shaft axis.

$\beta$  Flow inclination.

$\delta$  Wind turbine tilt angle.

$F_1$  Calibration factor mainly related to wind speed calibration.

$F_\alpha$  Calibration factor related to angle calibration.

$\gamma$  Yaw misalignment.

$\phi$  Rotor azimuth position (equal to zero when sonic sensor 1 is at top position, positive clockwise seen from the front of the wind turbine.

$\phi_s$  Angle of the sensor path respect to the mounting plate.

$k_1$  Calibration constant mainly related to wind speed calibration.

$k_\alpha$  Calibration constant related to angle calibration.

$k_2$  Calibration constant (equal to $k_\alpha \cdot k_1$).

$m$  Slope coefficient of the wind tunnel calibration equation (generic).

$q$  Offset of the wind tunnel calibration equation (generic).

$R$  Radius of the sphere approximating the pinner.

$r$  Radial coordinate in a polar coordinate system.

$\theta$  Azimuth position of the flow stagnation point on the spinner measured clockwise from sensor 1.

$U$  Wind speed vector modulus ($U = \sqrt{U_{hor}^2 + w^2}$).

$U_{hor}$  Horizontal wind speed (calibrated).

$U_{hor,d}$  Horizontal wind speed (measured with default values $k_{1,d}$ and $k_{2,d}$).

$U_{hor,d,c}$  Horizontal wind speed (calibrated with correct $k_\alpha$ but not correct $k_1$).

$U_{mm}$  Horizontal wind speed measured by the met-mast at hub height.

$U_{mm,n}$  Horizontal wind speed measured by the met-mast at hub height, corrected to standard air density.



$U_{free4}$  Free wind speed calculated with the nacelle transfer function from spinner anemometer measurements (turbine 4).

$U_{free5}$  Free wind speed calculated with the nacelle transfer function from spinner anemometer measurements (turbine 5).

$U_0$  Free stream inlet wind speed used in the potential flow analysis.

$u_1$  Uncertainty on $V_1$.

$u_t$  Uncertainty on $V_t$.

$u_m$  Uncertainty on wind speed due to mounting imperfections.

$u_M$  Uncertainty due to the NTF method (seasonal variations equal to 0.02 $V_i$).

$u_{ME}$  Uncertainty to account for the discrepancies between different MEASNET wind tunnels.

$u_{mm}$  Uncertainty on $U_{mm}$.

$u_{sa4}$  Uncertainty on wind speed measurements of the spinner anemometer mounted on turbine T4.

$u_{sa5}$  Uncertainty on wind speed measurements of the spinner anemometer mounted on turbine T5.

$V_1$  is the wind speed in the sensor path 1.

$V_2$  is the wind speed in the sensor path 2.

$V_3$  is the wind speed in the sensor path 3.

$V_{ave}$  Average wind speed of sonic sensors.

$V_{avg}$  Annual average wind speed used to calculate the wind speed probability distribution.

$V_i$  Center wind speed of bin $i$.

$v_r$  Velocity component along radius in a polar coordinate system.

$v_t$  Velocity component perpendicular to radius in a polar coordinate system.

$V_t$  Wind tunnel air speed.

$w$  Vertical wind component.



## Appendix B: List of abbreviations

Cp   Power coefficient of a wind turbine

IEC  International Electrotechnical Commission

NPC  Nacelle power curve

5    NTF  Nacelle transfer function

PC   Power curve

s. a.  Spinner anemometer

SN   Serial Number

T4   Turbine 4

10   T5   Turbine 5

*Acknowledgements.* Romo wind A/S is acknowledged for the good collaboration along the project and for the help to provide the spinner anemometer and power data of the Nørrekær Enge wind farm. Vattenfall A/S is acknowledged for providing access to the wind turbines used in the test and the help for installation of the met-mast. This work was performed under the EUDP-2012-I project: iSpin (J.nr 64012-0107).





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
