# Peer review of "Nacelle power curve measurement with spinner anemometer and uncertainty evaluation"

_Wind Energy Science, 2016_

## Referee Comment (RC2)

**General remarks**

The methodology proposed provides excellent results for two wind turbines, one next to each other, operating in flat terrain. It seems that the authors expect similar results in complex terrain. It would reinforce the conclusions of the paper if the authors could provide some more arguments/evidences about the expected performance in complex terrain.

**Typos**

There are some typos:

Several times "pinner" is written instead of "spinner". Some "consist" instead of "consists" There is a "calculatio" instead of a "calculation" somewhere.

**Style recommendations**

I would write "wind turbine" instead of "turbine"

**1. Introduction**

"Measuring the performance..." by "Measuring the power performance..."

About the end of page 1. A comment should be included indicating that the flow in the spinner anemometer region is also disturbed, but the flow disturbance in this region can be modeled, in general, by simpler methods compared to the rotor-nacelle wake region case.

**2. Site description**

First paragraph, please indicate if your experimental setup complies with MEASNET standards.

**3. Spinner anemometer calibration**

Just before section 3.1: Although you explain what  $k_{\alpha}$  and  $k_1$  are in following pages, it is convenient to include an explanation sentence about these parameters just here, in order to facilitate the reading.

Please define  $U_{hor}$ ,  $\gamma$  and  $\beta$  just the first time they appear in the text. I recommend to do this with all your parameters.

"...path angle  $\phi$  was..." by "...path angle  $\phi_s$  was..."

**4. Application of the nacelle transfer function**

Figure 8: Please use the same symbols in the axes labels and in the text (for instance  $U_{mm}$  instead of Metmast, some other times your write Met-mast wind speed...). Please use a common labeling in all your figures because this facilitates a lot the reading and the analysis of your results. Take a look to all your figures having this suggestion in mind. I would define a proper symbol for your normalizing wind speed value, let us say  $U_C$ , and label  $U_{mm}U_C^{-1}$  [-]...

**8. Power curve**

With respect to figure 10. Although the error in AEP is the best indicator on the accuracy of your method, perhaps you could include in your figure 10 the difference lines in % with respect to a reference power in order to clearly see the difference between power curves at each velocity.

Figure 10. Is really power or  $P/P_R$ ?

**9.1 Uncertainty related to wind tunnel calibration of sonic sensors**

"The standard uncertainty  $(k_c = 1)$  was calculated by dividing the value reported in the certificates by two"...perhaps I am going old but I do not see why your consider one half of the uncertainty.

**9.2 Evaluation of spinner anemometer mounting**

"The positions of the sonic sensors on the spinner were measured in the plane of the photos..." I wonder if there is any frame distortion. I am used to PIV analysis and frame distortions mainly at the borders of the frames is an issue. Perhaps your could include a comment on this.

**9.3 Uncertainty in wind speed measurements due to the mounting imperfections**

"...the flow field was calculated for a mesh of 0.01.." by "...the flow field was calculated for a mesh of 0.01 m.."

**9.4 Combination of uncertainties through the spinner anemometer conversion algorithm**

RHS of expression (24) I guess you must change " $\frac{u_1}{\sqrt{3}}$ " by " $\frac{u_1}{3}$ "

"...the derivative was computed numerically." Most probably I am missing something, but it seems that the derivative can be done analytically. Just for information, do you do it numerically because to do it analytically is tedious or because there is not an analytic derivative?

"The uncertainty on  $k_a$ ..." by "The uncertainty on  $k_{\alpha}$ ..."

**10 Results of uncertainty analysis**

"The mounting accuracy was investigated by overlaying six photos of the spinner taken from ground level during rotation, each showing the corresponding sensor when it is at the side of the spinner. The photos unveil deviations in the order of  $\pm 2$  cm between some of the sensors" is a repetition.

**11 Discussion**

"In complex terrain, a spinner anemometer should be assigned the calibration and NTF measured on a identical wind turbine in a flat terrain. The free wind speed calculated applying the NTF to the spinner anemometer measurements in complex terrain will provide a free wind equivalent to the one of a flat site, with no need for site calibration" You are proving (with excellent results) your method with two wind turbines, one next to the other, in flat terrain. You should provide some more evidences/arguments about the performance of your methodology in complex terrain.

**References**

"Demurtas, G., Pedersen, T. F., and Zahle, F.: Calibration of a spinner anemometer for wind speed measurements, Wind Energy, 2016. Faith, A, F. and Morrison, A.: An Introduction to Fluid Mechanics, Cambridge University Press, pages 656-658, 2013." by "Demurtas, G., Pedersen, T. F., and Zahle, F.: Calibration of a spinner anemometer for wind speed measurements, Wind Energy, 2016. Faith, A, F. and Morrison, A.: An Introduction to Fluid Mechanics, Cambridge University Press, 2013.".

---

## Referee Comment (RC1) · Anonymous Referee #1 · 29 Sep 2016

This paper investigates the methods to evaluate the power performance of two wind turbines using spinner anemometers, where power curves of two adjacent wind turbines are measured by means of a common traceable calibrated met-mast and spinner on each turbine. It verifies the feasibility of using the spinner anemometer calibration and nacelle transfer function determined on a reference turbine. It is concluded that power performance measurement with use of spinner anemometer can be made within 0.38% difference in AEP for annual mean wind speed of 8 m/s.

This paper is organized in a good structure, and some following concerns are needed to be modified and clarified:

- 1. Last sentence in abstract: 0.10% and 0,38%. Please keep them uniformed.
- 2. Figure 5: why power (pu) not reach 1 when the wind speed (pu) reaches 1 ? Maybe

add unit for wind speed, otherwise delete the brackets.

3. Figure 10: is the unit of y-axis kW? Please make sure it is a pu value or not.

---

## Short Comment (SC1) · 3 Oct 2016

Dear authors,

After reading the paper submitted for publication in WES, I would like to take the opportunity of the open discussion to highlight a few points that I believe require more consideration. It would also be interesting to have an interactive discussion with other members of the "wind research community".

The comments concern the uncertainty assessment of the spinner anemometer:

1) On coverage factors:

In general, it should be made clearer in the paper at which coverage factors are the uncertainty values given. I assume standard uncertainties were used (as defined by

,

10-11 and p20 lines 12-13). In particular in section 9.4: - uncertainties u1, u2, u3 are most likely fully correlated as originating from the same wind tunnel, same procedure, etc etc. Hence, u\_ave would be = to u\_1. - same for u\_alpha. I would recomment considering u1,u2,u3 correlated with a correlation coefficient =1, and k\_alpha uncorrelated with the sensor velocities. - u\_U may thus be much larger (or lower)

Asssuming a uniform distribution is only acceptable when there is 0 probability that the uncertainty falls outside some specific bounds. Even in such a case, it might be more likely that the uncertainty falls at one part (center?) of the distribution than another (e.g. triangle, truncated normal, etc). The default distribution in the GUM is the Normal one. Practically, it must be stated that a uniform (or rectangular) distribution is assumed every time the standard uncertainty is divided by sqrt(3) in order to account for the uniform distribution. For example, in eq. 20, I do not see why the sensor wind velocity could not fall outside of the [a- a+] range. The simple flow model has some level of inadequacy that may imply tails in the distribution, outside of this range. Statement lines 10-12 p. 19 may also be discussed.

In any uncertainty assessment procedure, the correlation between uncertainty distributions may have a large impact on the total combined uncertainty. The degree of correlation is often hard to quantify and thus needs to be "guessed". In the entire

Eq. 10 should formally be called the "law of propagation of uncertainties" (p15, lines

**2) On uncertainty distributions**

3) On correlation between uncertainty components

paper, correlation was disregarded and eq. 10 used.

GUM, i.e. coverage factor k = 1 for a Normal distribution of uncertainties). If so, then stating it once in the beginning of Sections 3 and 9 would be helpful. Whenever k is different from 1, then its value must be provided.

\_\_\_\_\_

WESD
Note: in Fig. 16, it is shown that u\_U is relatively small compared to the prevailing components related to the mounting and the NTF. Hence, the question of correlation may not affect dramatically the final results.

Regards,

Antoine Borraccino

---

## Referee Comment (RC3) · Anonymous Referee #3 · 25 Oct 2016

**General Comments**

The manuscript addresses a topic of great relevance for the wind energy community by evaluation of the feasibility of spinner anemometer measurements for the measurement of wind turbine power curves. Moreover, the question of using multiple spinner anemometers without requiring individual measurements of a transfer function against a met mast are of practical relevance. The choice of the annual energy production calculated based on the measured power curves constitutes a reasonable measure for the usability and it is methodically sound. The results indicate an excellent agreement of the measured (and corrected) power curves obtained for two adjacent wind turbines at a flat terrain site.

However, the calibration and correction methods used for the spinner anemometer calibration and the derivation of the nacelle transfer function are only sparsely explained and the reader is not provided with all information to follow the procedure without additional reading or in-depth knowledge of the particular spinner anemometer. This concerns particularly the spinner anemometer calibration and data conversion.

Moreover, the conclusion, that the spinner anemometer may perform equally well in complex terrain, is drawn. This hypothesis needs additional commenting and arguments to support it, since the presented study did not incorporated any effects of turbulence intensity or wind veer on the NPC.

**Specific Comments**

**1. Introduction**

- The "list of objectives" (p.2, line 5ff) is rather a list of steps performed to achieve the goal of comparing the measuring techniques and the presented correction scheme.

**3. Spinner anemometer calibration**

- The calibration of the spinner anemometer is described only very sparsely. It is impossible to understand the procedure without briefly explaining the "$k_\alpha$ calibration" and "$k_1$ calibration" and distinguish them from the wind tunnel calibration coefficients. Although some references are given, at least an equation of the calibration should be presented to enable the reader to follow the calibration.

- It is not explained, why the wind tunnel calibration was not introduced in the conversion box as demanded by the authors. The difference between $\Phi_s$ and $\Phi$ remains unclear as well as the reason why the path angle of the calibration is not used. What is the benefit of describing something, that was not done?

- Which meaning or use have the averaged slope, offset and angle given in Table 1?

- The authors do not define, what they consider "good wind conditions" (p.5, line 3).

**4. Measurement database, data filtering and corrections**

- Is the wind direction condition based on 10-min averages (p.7, line 2)?

- Could you explain, why the four outliers were excluded (p.7, line 6ff) although it deviates from the IEC standard?

- Regarding the calibration coefficients (slope m, offset q) of the two sets of sonic sensors, the authors state, that the sensor on T5 has higher wind speed readings due to the smaller slope and smaller offset. This appears illogic, since both sensors should read the correct reference wind speed, when their respective calibration coefficients from the and tunnel are applied. If the authors are referring to the uncorrected, uncalibrated sonic anemometer signals as e.g. $U_{5,original}$, they should point this out. In this case, they should explain, why they do not simply use the calibration coefficients obtained in the wind tunnel calibration. Besides this, neither $U_5$ nor $U_{5,original}$ are defined.

- The normalization of the wind speeds is based on an unknown reference wind speed (p.7, line 15f). It should be clarified, if this wind speed is the same for all presented results.

- The quantities in Eq.(4) are neither introduced in the text nor in the list of used symbols. In general, the readability of the manuscript would be greatly enhanced, if the symbols would be introduced in the text upon first use instead of relying on the list of symbols only. At least a reference to the list of symbols in the Appendix is required in the introduction of the manuscript.

- The used abbreviations for the power P4, P4n etc. in Figure 5 are not introduced and not defined in the Appendix. Label consistency should be checked.

**5. Nacelle Transfer function measurement**

- A definition of the different wind speeds used from the spinner anemometer would improve the description of the compared quantities. What is the relation between $U_{5,original}$, $U_5$, $U_{hor}$ and $U_{free}$? Which corrections and factors are included?

- What is the reason for not using the air density correction in the NTF (p.9, line 1f)?

- From Figure 6, one can only conclude, that the claimed proportionality factor is 1 for wind speeds higher than the wind speed normalized to (which is not know).

**6. NTF self consistency check**

- Technically, the NPC fails the test at approx. 0.7 $U_{mm}$ in figure 7. A comment on this would be good. Labels need to be checked. The legend for the blue and red thresholds is cryptic.

**7. Application of the NTF**

- The first line of the Section repeats the last paragraph of the previous one.

- A reference to 1.2*rated wind speed as a condition for a match between spinner and met mast wind speed is given in p.11, line 13ff. It has not been mentioned before and if this is the condition used to select the wind speed for normalization, it should be noted.

**8. Power curves and AEP**

- Using the introduced variables could help to distinguish between the calculated and measured free wind speed referred to in p.12, line 1.

- Eqs. (5) and (6) are redundant. A general formulation of the air density correction is preferable.

**9. Uncertainty analysis**

- No reference is given for the "spinner anemometer conversion algorithm" on which the uncertainty of the horizontal wind speed is based.

- In Eq. (8) quantities a+ and a- are not defined and used for different meanings later on. Values of the used quantities are usually not inserted into the equations, but only the result is given.

- The combined uncertainty stated in Eq.(10) is missing the square of the partial derivative, as is Eq.(25)

- The color coded velocity in Fig. 13 seems to be the normalized wind speed $U/U_{ref}$ rather than the relative wind speed $U - U_{ref}$.

- Eq. (18) uses numbers instead of the general description with symbols.

- $U_{sa4}$ is not defined. Is it $U_{free4}$ or the uncertainty $u_{sa4}$?

**Technical Comments**

The layout and labeling of the plots should be checked for consistency. Several different labels for the same quantity are used, e.g. "Met-mast wind speed $(U_{mm})$[m/s]", "Met-mast $(U_{mm})$[ ]", "Met-mast [ ]", "Met-mast wind speed [ ]"...

p.3, fig.1:
The figure would benefit from additionally marking the turbines T4 and T5 in the sketch (right).

p.4, line 4ff:
The content of the last paragraph is repetitive and should be modified to only give the necessary information once.

p.6, line 12:
The word "increase" lacks an "s". A comma after "(Fig.4)" should be added.

p.9, line 4:
"Approximately 1:1" seems a quite fuzzy expression to me, which could be improved.

p.10, line 4:
"make" appears twice

p.10, line 10:
"utmost" is not very fitting. "outmost" might be the word.

p.10, line 12:
"trough" should be "through"

p.11 line 2:
The reference to Fig. 9 is before referring to Fig. 8.

p.11, Fig 8:
The values of the coefficient of determination $R^2$ should be given in the text, too. The y should be rounded to significant digits.

p.12, Fig 9:
The Figure caption lacks a description of the four different plots.

p.13, line 4f:
The sentence lacks a verb or the grammar should be checked.

p.13, lines 5ff:
Both sentences are confusingly structured. It should be "PC4 compares better with NPC4..:" rather than "PC4 with NPC4 compares better..."

p.18 line 1:
"Spinner" lacks the initial "s"

p.29 line 10:
"path with respect to"

p.29 line 16:
"spinner" instead of "pinner"

---

## Author Comment (AC1)

This paper investigates the methods to evaluate the power performance of two wind turbines using spinner anemometers, where power curves of two adjacent wind turbines are measured by means of a common traceable calibrated met-mast and spinner on each turbine. It verifies the feasibility of using the spinner anemometer calibration and nacelle transfer function determined on a reference turbine. It is concluded that power performance measurement with use of spinner anemometer can be made within 0.38% difference in AEP for annual mean wind speed of 8 m/s.

This paper is organized in a good structure, and some following concerns are needed to be modified and clarified:

1. Last sentence in abstract: 0.10% and 0,38%. Please keep them uniformed.
[Figure]

2. Figure 5: why power (pu) not reach 1 when the wind speed (pu) reaches 1 ? Maybe

add unit for wind speed, otherwise delete the brackets.
[Figure]

3. Figure 10: is the unit of y-axis kW ? Please make sure it is a pu value or not.

---

## Author Comment (AC2)

Dear authors,

After reading the paper submitted for publication in WES, I would like to take the opportunity of the open discussion to highlight a few points that I believe require more consideration. It would also be interesting to have an interactive discussion with other members of the "wind research community".

The comments concern the uncertainty assessment of the spinner anemometer:

1) On coverage factors:

In general, it should be made clearer in the paper at which coverage factors are the uncertainty values given. I assume standard uncertainties were used (as defined by

[Figure]

GUM, i.e. coverage factor k = 1 for a Normal distribution of uncertainties). If so, then stating it once in the beginning of Sections 3 and 9 would be helpful. Whenever k is different from 1, then its value must be provided.

2) On uncertainty distributions

Asssuming a uniform distribution is only acceptable when there is 0 probability that the uncertainty falls outside some specific bounds. Even in such a case, it might be more likely that the uncertainty falls at one part (center?) of the distribution than another (e.g. triangle, truncated normal, etc). The default distribution in the GUM is the Normal one. Practically, it must be stated that a uniform (or rectangular) distribution is assumed every time the standard uncertainty is divided by sqrt(3) in order to account for the uniform distribution. For example, in eq. 20, I do not see why the sensor wind velocity could not fall outside of the [a- a+] range. The simple flow model has some level of inadequacy that may imply tails in the distribution, outside of this range. Statement lines 10-12 p. 19 may also be discussed.

3) On correlation between uncertainty components

In any uncertainty assessment procedure, the correlation between uncertainty distributions may have a large impact on the total combined uncertainty. The degree of correlation is often hard to quantify and thus needs to be "guessed". In the entire paper, correlation was disregarded and eq. 10 used.

Eq. 10 should formally be called the "law of propagation of uncertainties" (p15, lines 10-11 and p20 lines 12-13).

In particular in section 9.4: - uncertainties u1, u2, u3 are most likely fully correlated as originating from the same wind tunnel, same procedure, etc etc. Hence, u_ave would be = to u_1. - same for u_alpha. I would recomment considering u1,u2,u3 correlated with a correlation coefficient =1, and k_alpha uncorrelated with the sensor velocities. - u_U may thus be much larger (or lower)

[Figure]

Note: in Fig. 16, it is shown that u_U is relatively small compared to the prevailing components related to the mounting and the NTF. Hence, the question of correlation may not affect dramatically the final results.

Regards,

Antoine Borraccino
* * *